# SWOT and Root Cause Analyses of Antimicrobial Resistance to Oral Antimicrobial Treatment of Cystitis

**DOI:** 10.3390/antibiotics13040328

**Published:** 2024-04-04

**Authors:** Pradeep Tyagi, Shachi Tyagi, Laurence Stewart, Scott Glickman

**Affiliations:** 1Department of Urology, University of Pittsburgh, Pittsburgh, PA 15213, USA; 2Division of Geriatric Medicine, University of Pittsburgh, Pittsburgh, PA 15213, USA; tyagis@upmc.edu; 3Spire Murrayfield Hospital, Edinburgh EH12 6UD, UK; laurence.stewart@edinburghurologyspecialists.com; 4Uropharma Ltd., Norwich NR1 1PY, UK; scott@uropharma.co.uk

**Keywords:** UTI, AMR, oral treatment and intravesical treatment

## Abstract

Nearly 150 million cases of urinary tract infections (UTIs) are reported each year, of which uncomplicated cystitis triggers > 25% of outpatient prescriptions of oral antimicrobial treatment (OAT). OAT aids immune cells infiltrating the urothelium in eliminating uropathogens capable of invading the urothelium and surviving hyperosmotic urine. This self-evident adaptability of uropathogens and the short interval between the introduction of Penicillin and the first report of antimicrobial resistance (AMR) implicate AMR as an evolutionary conserved heritable trait of mutant strains selected by the Darwinian principle to survive environmental threats through exponential proliferation. Therefore, AMR can only be countered by antimicrobial stewardship (AMS) following the principle of the five Ds—drug, dose, duration, drug route, and de-escalation. While convenient to administer, the onset of the minimum inhibitory concentration (MIC) for OAT in urine leaves a window of opportunity for uropathogens to survive the first contact with an antimicrobial and arm their descendant colonies with AMR for surviving subsequent higher urine antimicrobial levels. Meanwhile, the initial dose of intravesical antimicrobial treatment (IAT) may be well above the MIC. Therefore, the widespread clinical use of OAT for cystitis warrants an analysis of the strengths, weaknesses, opportunity, and threats (SWOTs) and a root cause analysis of the AMR associated with OAT and IAT.

## 1. Introduction

The earliest imprint of life left by microbes on Earth goes back by 3.8 billion years [1,2]. As the estimated age of our planet is only 4.54 billion years, those earliest imprints of life preserved on rocks are a testament to the adaptability of microbes [3] to survive the catastrophes [4] that made many larger life forms extinct. While their size is microscopic, an unparalleled proliferation pace with genetic drift [5,6,7] equips microbes to be the most prevalent life forms on the planet, capable of thriving amidst all other forms of life, including in the highly acidic pH of the stomach [3] and in hyperosmotic urine [8] of mammals. The variable pH and the osmolality of stored urine [9] nearly mimic the threats to survival faced by the ancestors of uropathogens surviving in hot springs, billions of years ago [1,2]. As illustrated in Figure 1, uropathogens physically attach and invade the bladder epithelium—called urothelium—to cause uncomplicated cystitis, which refers to a microbial infection restricted to the bladder of afebrile, immunocompetent, non-pregnant females which can be managed in an outpatient setting with first-line antibiotics; not associated with the risk of treatment failure from foci of bacterial colonies associated with urinary tract stones, obstruction, catheters, pyelonephritis, and urosepsis [10] and without any long-term sequela of recurrent infections. In contrast, complicated cystitis generally refers to the infection of the urothelium in males or infection together with obstruction, hydronephrosis, renal tract calculi, colovesical fistula, or a compromised immune system. While described by the symptoms of frequency, urgency, and/or dysuria [11], the overlap of these symptoms with interstitial cystitis symptoms and other urological diseases complicates the diagnosis of uncomplicated bacterial cystitis [12], hereafter referred to as cystitis in this paper. 

The infection of the urothelium can only occur after uropathogens successfully compete for available nutrients with the commensal microbiome [13] by employing mechanisms of commensalism [14], mutualism, and parasitism [15,16]. Many groups have suggested that the biochemical mechanisms driving the adaptability of uropathogens in the variable pH and osmolality of urine [17,18] are as dogged as the signaling pathways driving the gain of virulence [19,20,21] and antimicrobial resistance (AMR) [22]. From a clinical perspective, AMR refers to a microbial phenotype capable of surviving the exposure to a drug treatment course at a concentration anticipated to prevent its survival [20]. While the efficacy of antimicrobials for treating systemic infections (bacteriemia) can be reliably indexed by the plasma levels of antimicrobials above the minimum inhibitory concentration (MIC), the killing of the uropathogens causing cystitis is determined by urine levels above the MIC [23,24,25,26]. As illustrated by over 40-year-old studies on oral [27] and intravenous Fosfomycin [28], the delay in oral absorption and urinary excretion leads to a significant delay in urinary MIC in the treatment of uncomplicated cystitis [29,30] and could foster AMR development, which has not received adequate attention.

While there has been a lot of discussion on the role of the genetic [21,22] and phenotypic attributes of adaptable uropathogens [31] in AMR development, this review is focused on oral antimicrobial treatments (OATs), specifically their interaction with the human body (pharmacokinetics), delaying the achievement of MIC in urine [32,33], and the potential consequences. Uropathogens exposed to below-MIC levels in urine bestow AMR genes to descendant colonies for gaining an asymmetric advantage in combating antimicrobials [6,7]. Accordingly, this broad-based review focusses on the relevant published literature of the last two-to-three decades for describing the nuances of renal physiology [26,34,35] and pharmacokinetics relevant to understanding the strengths, weaknesses, opportunity, and threats (SWOT) of OAT in treatment successes and failures and recurrent infections [36]. The SWOT analysis of OAT is brought into sharper focus by the root cause analysis of the efficacy determinants-urine levels above the MIC at the first contact with uropathogens [37]—to benchmark AMR associated with OAT [8,38,39] to AMR associated with intravesical antimicrobial therapy (IAT). Accordingly, the selection process of articles was determined by the relevancy and the recency of urine pharmacokinetic information relevant for the SWOT and root cause analyses of AMR with or without antimicrobial stewardship (AMS) [40,41] practices in OAT for cystitis. It is important to undertake SWOT and root cause analyses of AMR in cystitis because AMR threatens to undermine the future of human healthcare as we understand it today. 

## 2. Antimicrobial Resistance (AMR)

The WHO now recognizes AMR as one of the greatest threats to global healthcare. The warning triggered by the WHO [20,39,42] within the clinical and scientific communities is captured in 249,067 results generated by a recent PubMed search on “Antimicrobial resistance”. The efforts to contain AMR have led to the rise in AMS within clinical governance oversight of prescriber practices as a contemporary feature within hospital services [4,43]. According to NHS England, AMS refers to an organizational or healthcare system-wide approach to retard the evolution of AMR by promoting and monitoring the judicious use of antimicrobials to preserve their future effectiveness by the optimal selection, dosing, and duration resulting in the best clinical outcome with minimal side effects to the patients [44]. If one looks at the menace of AMR through the lens of a geological time scale [4], it is easy to surmise that it is just an expression of the 3.8 billion-year-old evolutionary conserved hereditary trait [5,6,7,8,16] to adapt and survive environmental stresses such as natural or synthetic antimicrobial substances. Our inference is supported by the first published report on AMR arriving just a few years after the marketing of Penicillin [45]. 

### Contribution of Urinary Tract Infections (UTIs) to AMR

With ~150 million cases/year globally [36], UTI refers to an infection anywhere along the upper or lower urinary tract, but uncomplicated cystitis [11] is a term reserved for the infected urothelium of the bladder (Figure 1). An estimated 25–40% primary care prescriptions of OAT are triggered by uncomplicated cystitis [46,47]. As per the Infectious Diseases Society of America guidelines, first-line OAT for acute uncomplicated cystitis includes nitrofurantoin followed by sulfamethoxazole-trimethoprim, oral β-lactams, and fluoroquinolones should be reserved as the last line of defense against cystitis in outpatient settings [48]. The overuse of OAT was documented in 75% of uncomplicated cystitis cases from 2021 to 2022 in midwestern academic medical centers in the United States [49]. A larger sample size from the 2016 national Healthcare-Associated Infections prevalence survey in the UK found that the percentage of OAT overuse was 62.5% [50]. Therefore, it appears that non-compliance with professional society guidelines and AMS for cystitis are the likely major contributors to AMR [15,40,41], but other factors may also be involved, so further investigation is warranted. 

A multi-center observational cohort US study of 3779 patients recruited from a network of 15 geographically diverse emergency departments found comparable incidences of uncomplicated and complicated cystitis [38] triggered by common uropathogens: Gram-negative strains of *Escherichia coli* (63.2%), followed by *Klebsiella pneumoniae* (13.2%), *Proteus mirabilis*, Gram-positive strains of *Enterococcus faecalis* and other species (5.8%), and *Staphylococcus saprophyticus*. Patients with acute cystitis with prior exposure to OAT or injectable antimicrobials in the previous 90 days [38] had a mean odds ratio of 2.68 (95% confidence interval [CI]: 2.04–3.51) for isolating fluoroquinolone-resistant uropathogenic *E. coli* (UPEC) from their urine samples, whereas a history of multi-drug resistant (MDR) uropathogens showed a mean odds ratio of 6.93 (95% CI: 4.95–9.70). The incidence of fluoroquinolone-resistance in isolated UPEC ranged from 10.5% to 29.7% by site and uropathogens producing extended-spectrum β-lactamase ranged from 3.6% to 11.6% by site. Intriguingly, a third of patients with a resistant isolate did not have any documented risk factors for resistance [38]. Given that uropathogens endowed with specific virulence factors are also evidently surviving in hyperosmotic urine [17,18], it is likely that prior exposure to antibiotics give rise to microbial colonies with stimulated biochemical mechanisms that not only confers virulence but also predisposes to AMR [19,21] development.

## 3. The Urinary Bladder

The primary function of the bladder is not just to store the urine excreted by the kidneys but also to participate in the maintenance of the homeostasis of plasma osmolality by altering the pH and osmolality of urine [9,51,52,53,54]. The homeostasis of the plasma is hallmarked by the stability of the plasma pH, nutrient, and electrolyte levels. The bladder participates in homeostasis via endocrine and stretch-mediated urine reabsorption [9,51,52,54,55] to match with the constant-but-slow rate of urine production during sleep [56]. The multicellular layer of the urothelium [57] lining the bladder is a highly metabolically active [12] multifunctional tissue for monitoring bladder distension [58] and signaling constitutional changes in urine [53,59,60] to the detrusor, micturition centers, and the immune system [61]. The top layer of the urothelium relies on the restricted transcellular permeability [62] of the umbrella cells endowed with the asymmetric membrane on the apical side to survive the low pH and three-fold-higher osmotic pressure of urine [9,63] than the serum (Figure 1). The intermediate cells replacing the infected superficial urothelial cells are smaller in size than the mature umbrella cells [64], which leads to a greater passive paracellular diffusion of urine constituents during cystitis [65,66,67], which may account for a lower abdominal discomfort and dysuria, prominent symptoms of cystitis [12]. The highest prevalence of UPEC in human cystitis informed its selection for the inoculation of pig bladders to model human cystitis, where the ingress of UPEC into umbrella cells was noted near tight junctions (Figure 1) [68]. The ingress of uropathogens into the urothelium is resisted by the components of the innate immune system releasing matrix and chemicals that entrap the invading uropathogens, as illustrated in Figure 1.

It has been known for over forty years that host resistance [69] is a key determinant of cystitis prophylaxis. Major components of host responses are the neutrophils localized near the umbrella cells after infection by uropathogens (Figure 1). The neutrophils are the foot soldiers of the innate immune response [31,70]. Upon activation by chemokines [61], the neutrophils release mesh-like structures called neutrophil extracellular traps (NETs) to entrap the invading uropathogens in the urine-filled lumen [71] and those attaching to umbrella cells. The breakdown products of glycosaminoglycans products can mimic the function of NETs [72]. NETs include a DNA backbone embedded with antimicrobial proteins including myeloperoxidase, neutrophil elastase, and histones [73], which are detectable in urine dipsticks. The combative role of neutrophils and NETs can be inferred from the lower incidence of cystitis symptoms in the second and third trimesters of pregnancy, despite an abundance of asymptomatic bacteriuria (ASB) with uropathogens [39]. Pregnancy-related hormonal changes [74] modulate the innate immune cell activity and cause neutrophilia, marked by an abrupt rise in the number of neutrophils in the second and third trimesters of pregnancy [75]. Neutrophils are considered to modulate the innate and adaptive immune response [76] through interactions with macrophages, dendritic cells, natural killer cells, and T and B lymphocytes [77], generating reactive nitrogen species and reactive oxygen species. Neutrophils are chemoattracted by chemokines CXCL-1 and CXCL-8 to the urothelium for combating cystitis [61]. While elevated urine levels of chemokines [78] released by the infected urothelium [61] are associated with cystitis, the OAT-associated decline in the urine levels of chemokines marks the resolution of cystitis. The breakdown products of glycosaminoglycans [72] from the kidneys to the bladder are also detectable in urine.

## 4. Cystitis Etiology and Treatment

A history of two or more episodes of bacterial cystitis is an independent risk factor [15,79] for recurrent cystitis, defined as ≥2 episodes in 6 months or ≥3 in a year. Compared to men, the higher incidence of cystitis in women is generally linked to their shorter urethra and infrequent voiding [69] that thins the urothelial barrier [12,62,80] for the proximal seeding of uropathogens from the adjacent cavities of the vagina and anus [81]. It is also likely that the periodic shedding of the urothelium, observed exclusively in the females of mammals [82,83,84] may increase the inherent predisposition to cystitis [12] but frequent shedding of urothelium may be lowering the incidence of urothelial carcinoma in women [85,86].

As the microbiome of the bladder is influenced by the microbiome of adjacent organs [87,88], it is plausible that the symbionts of other organs may become opportunist uropathogens, hallmarked by their attachment to umbrella cells (Figure 1) and the possession of specific virulence factors [89,90,91]. Uropathogens are characterized by short life spans and the rapid proliferation of mutant colonies with a vertical and horizontal transmission of genetic information [92,93]—plasmid transfer—an attribute which helped identify the DNA as the genetic material [94]. Cystitis often exacerbates co-morbidities and complicates the lives of people with other diseases. It is responsible for more non-elective hospital admissions in multiple sclerosis than any of its plethora of impairments [95]. 

### 4.1. SWOTs of Oral Antimicrobial Therapy (OAT) for Cystitis

The primary strengths of OAT are the simplicity and convenience of administration. The self-medication of OAT also does not burden doctors or nurses as is the case with the parenteral administration of antimicrobials. The primary weaknesses of OAT are inter-individual variability in the clinical outcomes and AMR owing to inter-individual variability in pharmacokinetics [25,29,30,96,97,98] and hepatobiliary vs. renal clearance [20] delaying the achievement of the minimum inhibitory concentration (MIC) in urine [23,87]. The oral bioavailability of OAT differs with race [32,99], food intake [97], inter-individual variability in the gastric emptying time [100], posture (supine or ambulatory) [101,102], and the general health status of the patient [103]. MIC is the lowest antimicrobial concentration required to prevent the visible growth of the test strain of a microbe after a definite incubation period under strictly controlled in vitro conditions [104]. 

It is noteworthy that the antimicrobial effect of OAT on cystitis takes effect only after the absorbed dose fraction runs the gauntlet of pharmacokinetics to be filtered from the plasma by the kidneys into urine (Figure 2) [20,23,26,30,32,35,87,105]. The variability in the pharmacokinetics of OAT stems from variable absorption of drug from the gut [23,106], first pass metabolism of drug in the liver before its systemic distribution and renal excretion (Figure 2) for an intermittent ureteric delivery of OAT into urine. 

The projected plasma and urine plots in Figure 2 represent published human urine and plasma levels of nitrofurantoin in healthy human volunteers after oral dosing [23], which must be read with the following caveats: while there can be fixed sampling time points for plasma, there is no fixed sampling of urine after oral dosing. Based on the frequent voiding of patients with cystitis, one can infer that the concentration build-up reported for healthy volunteers [23] may not materialize in patients with cystitis as their frequent voiding prevents concentration buildup (Figure 2). Moreover, kidney function as indexed by creatinine clearance [24,32,34,35,107] exerts a critical impact on the urine levels of OAT to cause a characteristic delay in the MIC [28]. The delay in the urinary MIC [24,32,34,107] of OAT in the wake of the initial conditions (different from parenteral administration) is a threat for the emergence and prevalence of AMR, which can be countered by AMS practices. AMS represents an opportunity for the future sustained growth of newer OAT targeting resistant uropathogens. The threats to OAT stem from iatrogenic toxicity secondary to malaise and the dehydration of cystitis patients altering the urine flow rate [107] and pharmacokinetic parameters of OAT. Treatment failures can provoke cystitis to progress to pyelonephritis, urosepsis [10], and recurrent infections. Other threats are the activation of AMR either due to overuse [47,50,108] or environmental changes [4]. Therefore, the SWOTs of the OAT outlined in the chart below (Figure 3) can be briefly stated as follows: 

Strengths—simplicity and convenience of self-administration of fixed dosages and regimens facilitate adherence to patient-centered care away from hospitals.

Weaknesses—inter-individual variability in oral absorption [25,96,97] and hepatobiliary vs. renal clearance [20,26,35,105] is compounded into a larger inter-individual variability in clinical outcomes and AMR. 

Opportunities—predictable oral bioavailability [30,101,102,109] could reduce the delay in achieving urinary MIC [28,105], and informed clinical practices [110,111] could counter the emergence of AMR. 

Threats—Treatment failure [29] with AMR activation and *Clostridium difficile* infections [112] triggered by OAT prescribed for an extended duration [41,47,49,50] and iatrogenic renal toxicity in patients with oliguria, lower creatinine clearance [107], resulting in the overdosing of OAT [113], environmental changes accelerating AMR [4], and the development of non-antibiotic approaches [13,14,19,20,114,115,116,117].

#### Volume of Distribution (Vd) and Clearance of OAT

The difference in the clinical outcomes of Fosfomycin in burn victims [35] and of nitrofurantoin in pregnancy [118,119] can only be understood by the differences in the clearance brought about by the changes in the volume of distribution—a body weight-dependent pharmacokinetic parameter—which links observed plasma drug concentrations for a theoretical volume for the distribution (Vd) of any drug [120]. The counter-intuitive concept of Vd can also explain the huge divergence in the urine and plasma levels of OAT, which we are illustrating here with nitrofurantoin. Given that only 20–25% of the absorbed dose of nitrofurantoin is ultimately excreted in urine [23], the remainder of the absorbed dose (100 − 20 = 80)% should generate relatively higher plasma levels. On the contrary, 75–80% of the absorbed dose of nitrofurantoin (fraction not excreted in urine) is diluted in 0.46 L/kg [119] of body weight or 32.2 L for a 70 kg adult to generate a <1 mg/L plasma concentration, whereas 20–25% of an absorbed dose excreted by the kidneys is diluted in just 1.5 L of urine over 24 h to generate a concentration of >10 mg/L (Figure 2). Simply stated, the bigger denominator for a 75–80% dose leads to 25-fold-lower plasma levels than the urine levels of nitrofurantoin. 

### 4.2. Root Cause Analysis of AMR—Delay and Variability in Urinary MIC

The root cause analysis of AMR described here is limited to the factors within the intravesical space [25,113] that contribute to the determinations of antibiotic efficacy whereby OAT failures can provoke cystitis progression to pyelonephritis, urosepsis [10], and recurrent infections. The general factors that contribute to success of uropathogens, marked by AMR development rest with numbers and time, i.e., higher bacterial loads existing for longer time are likely to evolve AMR for dissemination [5,6,7]. 

Uropathogens can adapt to OAT through a panoply of pathways [15], including negative tropism, the acceleration of mutations [121], sheltering under biofilm [91], and expulsion of the drug by membrane pumps [111]. Biodiversity suggests that there may be other mechanisms as well [121]. To suppress multiple approaches driving the emergence of AMR in Gram-negative bacteria, pharmacokinetic/pharmacodynamic modelling of OAT [122] found that drug exposure varies for different drugs. The target plasma concentration needed to prevent AMR was estimated to be 4–1000-fold higher than the MIC [122], which is, however, unfeasible via the oral route because of the toxicity and the narrow therapeutic indices, which refer to the narrow margin between a toxic dose and dose needed for plasma MIC of OAT. The recommendation of the pharmacokinetic/pharmacodynamic modelling of OAT is consistent with the deployment of an overwhelming force upon the first contact with the adversary (microbes) for extinguishing any hope of resurgence by adversary (i.e., after gaining AMR) [123]. Accordingly, the urine concentration of OAT upon the first contact with the microbial colonies [6,7] infecting the urothelium is a crucial determinant of the emergence of a drug-resistant strain that proliferates exponentially to survive even subsequently higher drug concentration in urine (Figure 2). Hence, the duration (Figure 4) between the achievement of the first drug molecule in urine and urinary MIC is a proverbial “window of opportunity” for microbes to activate latent AMR genes [5,124] or shelter from OAT by entering into leaky tight junctions of the inflamed urothelium [65,68] or exfoliated vesicles suspended in urine. 

The figure above of a mirage graph vividly illustrates, that the OAT being dripped into urine of variable volumes at a variable pH from the kidneys to the bladder may sufficiently facilitate the evasive actions taken by microbes to gain AMR. The graph is referred to as a “mirage” because the data points are unknowable and indeterminable for a real-world treatment.

As depicted in the plots (Figure 2B), nitrofurantoin reaches its peak plasma concentration (C_max_) in the time range of 2–24 h after oral dosing [23], and only 20–25% of an absorbed dose of nitrofurantoin is ultimately excreted in urine [23], as opposed to the 40–60% excreted for sulfamethoxazole and trimethoprim [87,106]. Even though the peak urine concentration C_max_ of oral nitrofurantoin is twenty-five-fold higher than the plasma C_max_, the initial delay in reaching the urine MIC leaves a window of opportunity (Figure 3) for the activation of AMR genes to potentially survive subsequent higher concentrations of nitrofurantoin [5,6,7,108,125]. 

Since the human bladder [126,127] only receives 2% of the cardiac output, or one-tenth of the cardiac output delivered to the kidneys for producing urine [128,129], the exposure of bladder mucosa to the circulating drug levels [130] is minimal compared to the luminal exposure to the urine drug levels, as reported recently for the oral beta-3 agonist [127] (Figure 2). Simply stated, the metabolic demands for urine storage by bladder [126,127] are ten-fold lower than the metabolic demands for urine production. For all intents and purposes, a bladder’s exposure to OAT (Figure 3) leaves a time window of opportunity for the invading microbes to mutate and exponentially proliferate [5,6,7] owing to a large variability in the MIC delay for the following reasons: The variability in the initial drug concentration due to variable absorption from the gut [25,32,96,97,98,99,100,101,102,103], hepatobiliary vs. renal clearance delaying the achievement of the MIC in urine [20,23,28,30,87,101,105,109], and variability in urine in-flow rate of 0.3–15 mL/min (~50× difference) [26,56] could reduce the delay in achieving the urinary MIC;The antecedent intravesical antimicrobial-free urine volume [80] that can range from <10 mL to potentially well-over a liter (>100× difference) [131];The urinary pH physiological range of 4.4–9.9, a log-scale representing a >300,000-fold difference in acidity/alkalinity, a determinant of the aqueous solubility of the excreted drugs [107,132,133] and their reabsorption by the bladder [120].

A simple multiplication of the noted ranges (>50× > 100× > 300,000×) indicates that there are differences in the physical and physicochemical characteristics of urine [133], in stark contrast to the plasma volume and pH controlled by the homeostatic mechanisms for the optimal functioning of cells. As a result, the variability in the time taken to reach the peak urinary concentration (T_max_) of OAT results from the highly variable physiological and pharmacokinetic activities preceding the entry of OAT into urine, which are the following: Inter-individual differences in the metabolism [134];Fluid and electrolyte consumption, e.g., dietary restriction of sodium significantly increases 24 h voided urine volume [63];Inter-individual pharmacokinetic differences [122].

A study on healthy humans [23] indicated the role of the factors listed above in the inter-individual variability in the mean value of urine T_max,_ which is delayed by 2.5 h relative to the plasma T_max_. The variability in the urine T_max_ can potentially contribute to the variability in the emergence of AMR [8]. While the MIC of OAT is sensitive to the pH [107], the extrapolation of in vitro MIC to the bladder is rendered untenable by a wide range of potential intravesical acidity/alkalinity. It is noteworthy that the laboratory-based efficacy testing of the MIC for cystitis [135,136] does not account for the variability in urine flow, as higher urinary flow rates lead to a lower urinary MIC of Fosfomycin [24,26,133]. 

Furthermore, cystitis can dilate the tight junctions of the inflamed urothelium [12,65,82] to reportedly cause a four-fold rise in creatinine reabsorption [55,67] via passive paracellular diffusion, and the resulting decline in the urine creatinine levels can potentially introduce errors in the estimation of the renal function status of cystitis patients, leading to their under- and overdosing of OAT [25,113]. Given that every dose of an OAT regimen produces an initial subtherapeutic concentration in urine [122] during the intended build-up to the MIC, which is itself indeterminable, there is an inevitable period of potential AMR risk with the initiation of OAT, even with AMS [40,41,137,138,139]. Thus, our root cause analysis (studies listed in Table 1) identifies the initial delay in the MIC as the root cause of AMR, which is a risk with every treatment course of OAT for cystitis.

## 5. Discussion

It has been known for over forty years that decreased host resistance [69] is a key determinant of cystitis secondary to the disruption of tissue integrity [12,65,83] or a decrease in the blood supply to the bladder either due to psychogenic stress [82] or abdominal aortic calcification [141]. A recent study reported the non-inferiority of a new triazaacenaphthylene antibiotic, Gepotidacin, relative to nitrofurantoin in treating uncomplicated cystitis caused by UPEC and nitrofurantoin-resistant uropathogens [33]. Gepotidacin inhibits eukaryotic DNA replication by inhibiting DNA gyrase as well as topoisomerase IV for a multipronged action on uropathogens. 

Although newer antimicrobials arrive periodically on the market, drug development path through a regulatory maze cannot keep pace with the pace of AMR development by uropathogens. Accordingly, with the limited availability of treatment options for resistant uropathogens, many experts advise us to try behavioral modifications of healthy voiding regimens that avoid the overstretching of the bladder’s mucosal barrier by hydrostatic pressure [69,142]. OAT for uncomplicated cystitis should only be initiated if the dysuria is refractory to non-steroidal analgesics. OAT prescriptions [143] for asymptomatic bacteriuria (ASB) with pyuria [142] have the potential to alter the diversity of the microbiome in favor of resistant pathogens [38] and adversely impact the memory of innate immune cells to combat uropathogens. A widespread preference for oral fluoroquinolones to treat uncomplicated cystitis is certainly not in accordance with AMS [41] or the Infectious Disease Society of America guidelines and risks potential adverse reactions like *Clostridium difficile* infections [144]. Antibiotic-associated diarrhea (AAD) is a prominent side effect of the OAT prescribed for cystitis in the elderly which risks serious sequalae, including *Clostridium difficile* infections with longer durations of OAT [112].

A major contributor of nosocomial infections are urinary catheters and medical devices providing access to uropathogens and increasing healthcare expenditures. Nearly 70–80% of complicated UTI are attributable to catheter-associated urinary tract infections—CAUTI [31]—which are generally associated with the prolonged hospitalization of older females with or without diabetes. Over the last decade, transurethral catheters coated with novel surface modifications [145], including superhydrophilic zwitterionic surfaces, slippery liquid-infused porous surfaces, covalently attached liquid-like surfaces, and superhydrophobic surfaces [146], have been studied to retard the encrustation and colonization of biofilm-forming uropathogens. It has been shown that, while the antibiofilm coating of zinc oxide nanoparticles on silicone catheters could not withstand the corrosive effect of artificial urine for 14 days, a protective layer of carbon and silica oxide dramatically reduced the erosion of antibiofilm activity against Staphylococcus aureus in urine [147]. Given that periodic instillation of *Lactobacillus rhamnosus GG* [13] disrupted the colonization of urothelium by virulent strains of uropathogens in 26 self-catheterizing neurogenic bladder patients, the 3D bioprinting [148] of *Lactobacillus rhamnosus GG*-containing silicone scaffolds on catheters seems to be a promising strategy to prevent CAUTI. A recent pilot clinical study from China demonstrated the promise of blocking biofilm formation in critically ill patients with the use of a silver alloy hydrogel-coated catheter to deliver microcurrents to the adhering uropathogens [149].

While novel coated catheters could reduce the risk of complicated cystitis among inpatients, the management of uncomplicated cystitis can only improve with the guideline adherent antibiotic-prescribing behaviors, but adherence remains low across the world, with over 10% of OAT prescription for ASB in Australian hospitals [47]. A tidal wave of momentum recognizes AMS as a critical tool to assail AMR [150]. Last October, the National Health Service (NHS) of England reported that AMR has been associated with 1.8 million hospital admissions in the last 5 years and that UTI was causative in over 800,000 of them, with people over 65 years of age presenting the most cases. A quarter of the urine samples analyzed in the first half of 2023 had drug-resistant bacteria, and there was only a 1.6% reduction in the total number of antibiotic-resistant infections from 2018 to 2023, far below the ambitious goal of 10% by 2025. AMR is believed to be the proximate cause of over 300,000 deaths annually in India, the most populated country [42]. A systematic review of AMS in long-term care facilities in France found that the volume of antibiotic prescriptions was positively associated with staff turnover and after-hours medical practitioner visits, which could be corrected by an on-site coordinating physician [151]. Likewise, a multifaceted intervention for AMS in different parts of the world resulted in either reduction of treatment duration [152,153] or in better informed choice of OAT [137,138]. 

A multimodal AMS initiative decreased outpatient usage of fluoroquinolones for cystitis by 39% [40] in a large health system of Texas from 2016 to 2018, and a similar effort in Colorado, US, was more successful in encouraging the use of nitrofurantoin as a first-line therapy for acute uncomplicated cystitis [139]. The overuse of OAT was also noted in a retrospective study of Medicare Part B enrollees in the New York State between 2016 and 2017, which found that, although AMS efforts slightly reduced the use of oral fluoroquinolones for uncomplicated cystitis, there was still a need for additional outpatient AMS efforts to curtail the use of oral fluoroquinolones in American hospitals [41,49]. The value of AMS in cystitis can be better understood by understanding the critical determinants of AMR prevalence [38,42], which are current clinical practices of OAT prescriptions for cystitis and the physiology of urine flow in the bladder [9,51,52], as outlined below: (1)Cystitis treatment failures and AMR result from a bladder milieu that allows adaptable microbes [8] to shelter under a biofilm [91] to evade antimicrobials [38], and membrane pumps [121] allow them to expel OAT;(2)Interchangeable use of the term UTI and cystitis. While absorbed OAT can reach pathogens in upper urinary tracts via perfused blood, the uropathogens causing cystitis can only be eliminated by the urinary fraction of absorbed OAT (Figure 2), because the circulating levels [126,127,130] of the drug are less likely to reach the planktonic microbes and those attaching to the apical side of umbrella cells (Figure 1);(3)The battle between OAT and the uropathogens provoking cystitis is heavily influenced by the physiology of urine flow [20,25,113] and drug physiochemistry [107,132];(4)The principle of the five Ds—drug, dose, duration, drug route, and de-escalation—for AMS [44] advocates for obtaining the right drug concentration in urine above the MIC (Figure 2, Figure 3, Figure 4 and Figure 5) [124,154].

The principle of the five “Ds” can be best illustrated by an intravesical gentamicin dose of 80 mg [155,156,157,158,159] (Figure 5B), delivering a bactericidal concentration with a 90% elimination of live bacteria within 3 h of administration, with a comparative lower incidence of AMR [140] and recurrence [37] than those reported with OAT [38,123]. As opposed to the urine and plasma plots of OAT shown in Figure 2B, the plot in Figure 5B is projected from a recent study on post-operative prophylaxis by intravesical gentamicin [160]. As depicted by the low plasma levels (red curve in Figure 5B), the limited bioavailability of drugs from the bladder can allow the local administration of a drug concentration (yellow curve in Figure 5B) that would be in a toxic range if administered systemically by oral or intravenous route [154]. Accordingly, IAT gains an asymmetric advantage in combating adaptable uropathogens by delivering a concentration 4–1000-fold higher than the MIC [122], even biofilm-inhibitory and eradication concentrations [91], upon its first contact with microbes and thereby lowers the likelihood of AMR development [123], whereas OAT exposes the bladder urothelium to a higher concentration [23,106] in a gradual fashion, allowing uropathogens to adapt and survive with AMR [39,150]. Although IAT (Figure 5) complies with the prescription of the five “Ds”, IAT with aminoglycosides or any other broad-spectrum antibiotic, to our knowledge, is yet to obtain regulatory approval for therapy or prophylaxis [124]. However, a plethora of recent reports [91,155,156,157,158,159] demonstrate a widespread off-label use of aminoglycosides as an IAT for recurrent cystitis and prophylaxis. While gentamicin has been safely used in IAT, other aminoglycosides have been reported to cause perception deafness even after intravesical administration in patients with renal failure [161,162]. 

A major limitation of our study is that it is not an exhaustive review of the global threat of AMR, and this review is focused on the pharmacokinetic attributes of antimicrobials and not on the genomic attributes of the microbes in AMR development. The weakness and threats of OAT stemming from its poor bioavailability and renal insufficiency are thrown into sharp relief by the immediate delivery of antimicrobials via intravesical administration for the recurrent infection of uncomplicated cystitis. We performed a SWOT analysis of OAT here, but a SWOT analysis of the uropathogens may also be necessary to ensure treatment success, which is defined as either complete symptom resolution or a reduction in qualifying uropathogens to <10^3^ CFU/mL [33]. The major strength of uropathogens is their adaptability to survive hyperosmotic urine, while their weakness can be gleaned from the susceptibility reports, and the AMR threats emanate from both unknown and indeterminable T_max_ in urine, which are moving targets. As illustrated by Figure 2, the variability in the initial intravesical conditions [6,7] can be consequential to the OAT sensitivity of colonies descending from uropathogens exposed to the initial conditions [7]. However, standard OAT offers no indication of the duration of the window of opportunity. We suggest that cystitis should be treated as a distinct entity within the broad category of UTIs, in light of the SWOT and root cause analysis of AMR discussed here. While IAT delivers antimicrobials directly to the site of infection in cystitis (Figure 5), OAT reaches the same site after renal excretion [25,113], and, therefore, the dynamic entry of the excreted fraction of absorbed OAT in urine is a critical factor in the efficacy of OAT and AMR risk [20]. 

Since a key component of the host’s response against cystitis is the chemoattraction of neutrophils [163] releasing NETs to entrap uropathogens [76], many groups have tried to structurally mimic NETs for cystitis prophylaxis. Glycosaminoglycans mimic the entrapment of uropathogens by NETs [72], and a recent retrospective analysis of 151 patients found intravesical chondroitin sulphate to be superior to OAT in the treatment of recurrent cystitis [117]. This recent study confirmed reports from other groups using hyaluronic acid and chondroitin sulphate for the prophylaxis of cystitis in individuals with an injured spinal cord and in others with an intact spinal cord intact [114,116]. Other groups have explored antimicrobial peptides [20], bacteriophage treatments [19], and natural products [115] as non-antibiotic prophylactic approaches for UTIs. Analogous to the fecal microbiota transplantation for managing recurrent *Clostridioides difficile* infections [164], the intravesical instillation of innocuous microbes into the bladder [13,14] is a non-antibiotic approach for cystitis. The intravesical inoculation of the *E. coli* 83972 ASB strain in the atonic bladders of 21 patients with an injured spinal cord and chronic bacteriuria [14] achieved successful long-term bladder colonization in 13 of the participants, which lasted for more than a year without causing bacteremia or sepsis. 

Recent studies suggest that evolutionary changes may mitigate AMR [3], but, since hope is never a winning strategy in any war, we should be ready for AMR being aggravated by climate change [4]. To slow the pace of the evolution of AMR, the US Food and Drug Administration and the European Medicines Agency recently approved Dequalinium chloride to inhibit the activation of the *Escherichia coli* general stress response, which promotes fluoroquinolone-induced mutagenic DNA break repair [22]. Despite the fortuitous successful OAT of a high proportion of cases, this approach may produce failures and accentuate the evolution of AMR and the complications we observe in clinical services. The extrapolation of clinical experience in the treatment of bacterial infections in other parts of the body to cystitis should be re-evaluated in light of the pharmacokinetic considerations described here as an essential feature of AMS [110]. Based on the current understanding of the adaptability of uropathogens, we deign to hypothesize that any antibacterial treatment protocol [124] that fails to account for the evolutionary predisposition of microbes that have been adapting to climate change for billions of years [4] may be destined to fail the test of time. As the future of antibiotics evidently rests so firmly upon AMS [40,41,150], we leave this thought for all the stakeholders in health to consider. 

## 6. Conclusions

The SWOTs analysis of OAT reaffirms that the duration of the drug levels above and below the MIC at the site of cystitis is the primary determinant of the efficacy and AMR, respectively [122]. Therefore, we must give serious consideration to the AMR risks associated with OAT. 

## Figures and Tables

**Figure 1 antibiotics-13-00328-f001:**
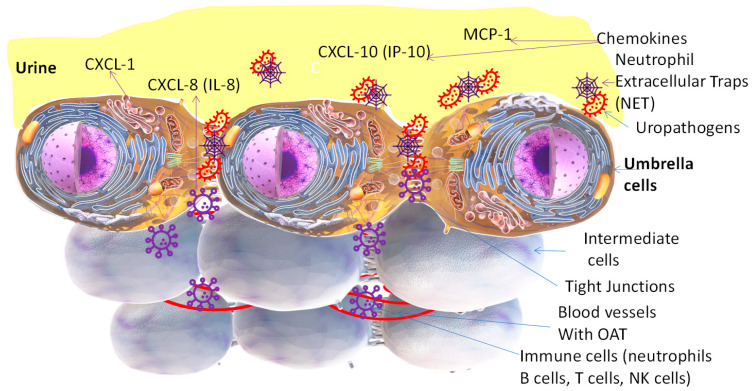
The bladder luminal surface is lined with long-lived, highly differentiated, hexagonal-shaped umbrella cells ranging 20–150 µm in diameter. Although asymmetric membrane on apical surface of umbrella cells is hallmarked by a restricted permeability to urine constituents, the apico-lateral surface between adjacent umbrella cell borders is lined by finger-like projections—tight junctions—amenable to the passive paracellular diffusion of urine constituents. Infiltrating neutrophils near the infected foci release neutrophil extracellular traps (NETs) into the bladder lumen to ensnare the invading uropathogens, and released leucocyte esterase is detected in urine dipstick tests for UTIs.

**Figure 2 antibiotics-13-00328-f002:**
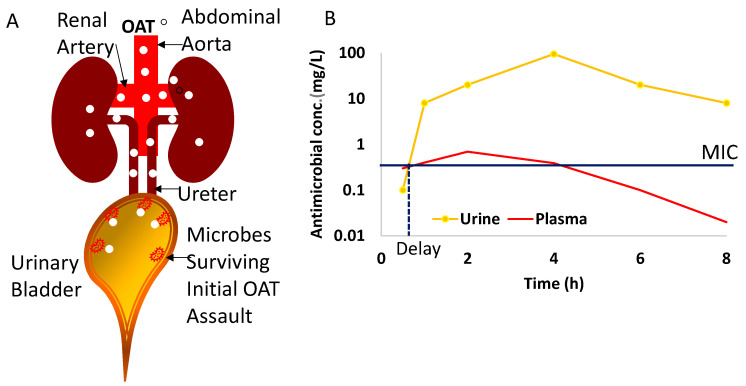
Panel (**A**)—Oral antimicrobial treatment (OAT) reaches the uropathogens infecting the bladder urothelium after the absorbed dose fraction is delivered to the kidneys via renal arteries and the OAT is excreted into urine. As a result, there is an initial mismatch between the initial urine concentration of OAT and the number of microbes infecting the urothelium. Panel (**B**)—Projected plots for serum and urine levels after a single dose of OAT (nitrofurantoin). Time 0 is the time of drug administration and the delay in reaching the urinary MIC leaves a window of opportunity for the activation of AMR genes. Then, the resistant descendants selected by the Darwinian principle proliferate exponentially. Please note the log-scale of the y-axis.

**Figure 3 antibiotics-13-00328-f003:**
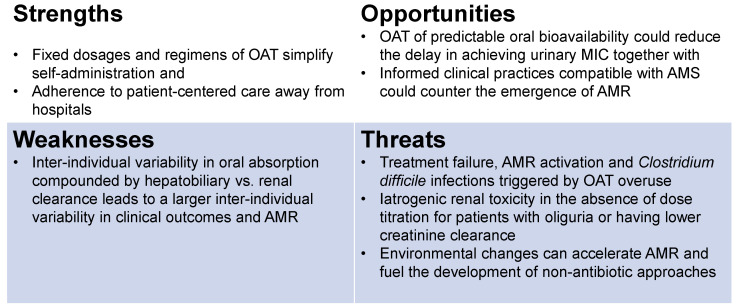
SWOT chart for OAT in uncomplicated cystitis: while OAT is simple and convenient, uropathogens forewarned by the delay in urinary MIC can pose an insurmountable challenge in resolution of cystitis.

**Figure 4 antibiotics-13-00328-f004:**
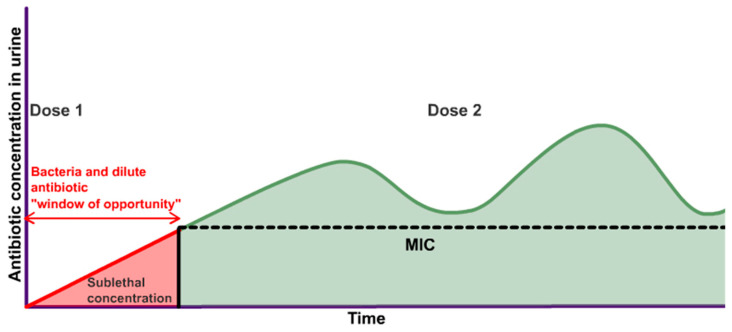
An illustrative graph for the repeat dosing of OAT is projected to engender time-dependent variability in antibiotic concentration in urine with respect to the MIC and leave a window of opportunity for the activation of AMR. The non-zero interval below the MIC for the red portion of the curve is in hours, but the exact duration remains indeterminable.

**Figure 5 antibiotics-13-00328-f005:**
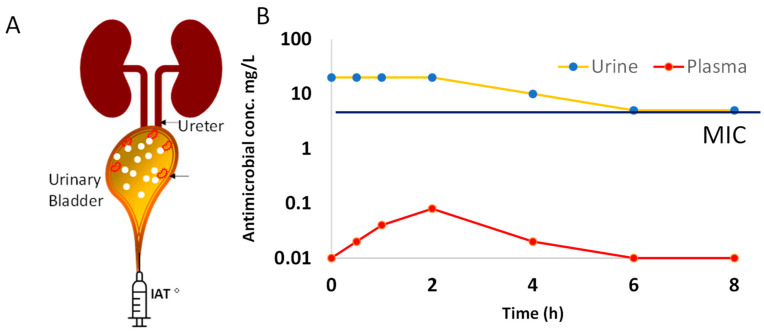
Panel (**A**)—Intravesical antimicrobial therapy (IAT) delivers a bactericidal dose to the microbial colonies invading the bladder without the delay associated with OAT. Exposure to the high dose does not give as much time for the activation of AMR and the proliferation of resistant colonies. Panel (**B**)—Projected plots for the serum and urine levels after a single intravesical dose of aminoglycoside (gentamicin) for cystitis reveal an absence of delay in reaching the MIC and delivering a lethal blow to the microbes, with concomitant low plasma levels, owing to the poor systemic absorption, avoiding renal and ototoxicity. Notice the log-scale of the y-axis.

**Table 1 antibiotics-13-00328-t001:** Key studies underpinning the essence of SWOTs and root cause analyses.

Studies	Key Findings/Implications
Wright et al., 2021 [6]	Study aids drawing the link between the delay in urinary MIC and time dependent evolution of uropathogens with AMR
Edwina et al., 2023 [24]	Delay in the attainment of urinary MIC of Fosfomycin relative the plasma MIC
Mponponsuo et al., 2023 [109]; Forsberg et al., 2023 [103]	Significance of oral bioavailability in clinical outcomes of OAT in uncomplicated cystitis
Faine et al., 2022 [38]	Prevalence of fluoroquinolone-resistant cystitis evinces a modest compliance with AMS and risks potential adverse reactions like *Clostridium difficile* infections
Cunha et al., 2016 [107]; Potel et al., 1989 [35]	Importance of renal insufficiency, urinary pH, and volume of distribution in therapeutic efficacy of OAT and injectable treatments
Sumi et al., 2019 [122]	OAT concentration required to suppress AMR
Andretta et al., 2022 [140]	Evidence supporting the efficacy of intravesical administration to counter AMR and recurrent cystitis

## Data Availability

No new data were created or analyzed in this study. Data sharing is not applicable to this article.

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
