# Peer review of "SWOT and Root Cause Analyses of Antimicrobial Resistance to Oral Antimicrobial Treatment of Cystitis"

_antibiotics, 2024, doi:10.3390/antibiotics13040328_

Round 1

Reviewer 1 Report (New Reviewer)

Comments and Suggestions for Authors

The Literature Review - SWOT and Root Cause Analysis of Antimicrobial Resistance to Oral Antimicrobial Treatments for Cystitis - addresses a timely subject in the medical realm. However, there are several suggested modifications:

1.       The introduction should be expanded to provide more comprehensive discussion on the pathology of cystitis.

2.       A concise Materials and Methods section should follow the introduction.

3.       The Discussion section requires improvement by incorporating a table summarizing key literature articles related to the topic. Additionally, it should cover antibiotic therapy in urological pathology, particularly focusing on potential adverse reactions like Clostridium difficile infections. More information can be found at https://doi.org/10.37358/RC.17.7.5694. Moreover, a brief overview of the evolution of materials in urology, resulting in enhanced efficacy in treating pathologies (especially those related to the bladder) and consequently reducing risks of infections, cystitis, hemorrhages, and severe conditions, should be included. Relevant literature sources such as https://revmaterialeplastice.ro/pdf/21%20GRIGORE%20N%20%202%2017.pdf can be consulted for reference.

4.       A Conclusions section should be added to augment the article's value.

5.       Please ensure the inclusion of all typical MDPI journal structures, such as Author Contributions, etc.

Comments on the Quality of English Language

Minor editing of English language required.

Author Response

  1. The introduction should be expanded to provide more comprehensive discussion on the pathology of cystitis.

Response: We thank the reviewer for the constructive suggestion, and we have now added a comprehensive discussion on the pathology of complicated and uncomplicated cystitis in the introduction section (Lines 40-52) to complement a separate section on the etiology preceding SWOT section in manuscript.

  1. A concise Materials and Methods section should follow the introduction.

Response: We would like to clarify that last paragraph of the introduction describes the rationale for the undertaking the review and the criteria used for selecting articles to perform SWOT of oral antimicrobial treatment and the root cause analysis of AMR (Lines 75-85).

  1. The Discussion section requires improvement by incorporating a table summarizing key literature articles related to the topic. Additionally, it should cover antibiotic therapy in urological pathology, particularly focusing on potential adverse reactions like Clostridium difficile infections. More information can be found at https://doi.org/10.37358/RC.17.7.5694. Moreover, a brief overview of the evolution of materials in urology, resulting in enhanced efficacy in treating pathologies (especially those related to the bladder) and consequently reducing risks of infections, cystitis, hemorrhages, and severe conditions, should be included. Relevant literature sources such as https://revmaterialeplastice.ro/pdf/21%20GRIGORE%20N%20%202%2017.pdf can be consulted for reference.

Response: We thank the reviewer for the insightful remark on Clostridium difficile infections. We have now incorporated a table summarizing key literature articles related to the topic in root cause analysis. We thank the reviewer for sharing reports on Clostridium difficile infections and we have now cited freely accessible reports available in PubMed central at lines 416-419 of manuscript.

  1. A Conclusions section should be added to augment the article's value.

Response: We thank the reviewer for the suggestion and we have now added a conclusion section to the manuscript.

  1. Please ensure the inclusion of all typical MDPI journal structures, such as Author Contributions, etc.

Response: We thank the reviewer for pointing that out and author contribution section is there at the end of manuscript.

Reviewer 2 Report (New Reviewer)

Comments and Suggestions for Authors

Congratulations on your efforts in concentrating so much information in the present article.

I have two remarks:

1.  The authors should include a Chart for the SWOT analysis. It would be more accessible to follow.

2. The authors also should explain the selection process for included articles.

3. The limitations of this study should be explained.

Author Response

Congratulations on your efforts in concentrating so much information in the present article.

Response: We highly appreciate the reviewer’s comments

I have two remarks:

  1. The authors should include a Chart for the SWOT analysis. It would be more accessible to follow.

Response: We thank the reviewer for the suggestion, and we have now included a chart of SWOT analysis in section 4.1 of the revised manuscript.  

  1. The authors also should explain the selection process for included articles.

Response: We thank the reviewer for the insightful remark. The selection process was determined by whether the published literature is recent and/or contains information relevant for SWOT and root cause analysis of AMR with oral antimicrobial treatment for cystitis. Accordingly, we prioritized articles containing urine pharmacokinetic of oral antimicrobial treatment or those reporting AMR with or without following antimicrobial stewardship practices. We have included this information in the introduction section (Lines 75-85).

  1. The limitations of this study should be explained.

Response: We thank the reviewer for the critical comment and limitations are now included in the discussion section of revised manuscript. This is not an exhaustive review on the global threat of AMR and the review is focused on the pharmacokinetic attributes of antimicrobial and not on the genomic attributes of microbe itself in AMR development (Lines 494-498).

Reviewer 3 Report (New Reviewer)

Comments and Suggestions for Authors

Overall this paper is interesting in looking into the problem of whether oral antimicrobial therapy (OAT) is sufficient for the treatment of cystitis. The authors proposed several interesting concepts including whether OAT is associated with a time lag in achieving urinary MIC for certain bacteria, treating cystitis as a separate entity of UTI, as well as the use of intravesical antibiotics in the treatment of cystitis.  I appreciate the authors' analysis of why the use of OAT is sometimes associated with the development of AMR, and different factors that may affect the urinary antibiotics concentration, therefore, leading to the emergence of resistance. I only have several minor comments concerning this version of the manuscript:

1. Line 110: Enterococcus faecalis and other species (No need to italicize "and other species")

2. Line 157: "3.1" is not necessary

3. Line 164 - 166: Please check whether the statement is accurate. Urine dipsticks usually include leucocyte esterase, but the mentioned proteins are not commonly included in urine dipsticks.

4. Line 208: the "arrival" of minimum inhibitory concentration (MIC) in urine - it might be better for the authors to consider the use of achievement instead of arrival.

5. Line 386 - 387: There should be some errors in terms of reference. Please double check.

6. Line 502: Escherichia coli should be italicized.

Author Response

I only have several minor comments concerning this version of the manuscript:

  1. Line 110: Enterococcus faecalis and other species (No need to italicize "and other species")

Response: We thank the reviewer for the encouraging remarks and pointing out the error. We have now fixed the error in revised manuscript.

  1. Line 157: "3.1" is not necessary

Response: We thank the reviewer and we have now deleted that heading.

  1. Line 164 - 166: Please check whether the statement is accurate. Urine dipsticks usually include leucocyte esterase, but the mentioned tproteins are not commonly included in urine dipsticks.

Response: We thank the reviewer for the insightful remark, and we have now clearly specified the leucocyte esterase in revised manuscript.

  1. Line 208: the "arrival" of minimum inhibitory concentration (MIC) in urine - it might be better for the authors to consider the use of achievement instead of arrival.

Response: We thank the reviewer for the insightful remark, and we have now made the correction at multiple places in revised manuscript.

  1. Line 386 - 387: There should be some errors in terms of reference. Please double check.

Response: We thank the reviewer for the comment, and we have now fixed the reference.

  1. Line 502: Escherichia coli should be italicized.

Response: We thank the reviewer for the comment, and E. coli is now italicized in revised manuscript

Reviewer 4 Report (New Reviewer)

Comments and Suggestions for Authors

This is a very interesting well written manuscript discussing the most appropriate way of antibiotic treatment modality  of cystitis. If I understand it correctly, the principle idea of the pharmacokinetic consideration is that sufficient antibacterial urinary concentrations in the bladder always will take some time after oral treatment of whatever antibiotic. Therefore, there will be always some time with subinhibitory and/or subbactericidal concentrations. In this situation the pathogens – some of them even localized intraepithelially – may develop resistance to the antibiotic administered. This general consideration is of course interesting. The authors support this idea by several molecular biological mechanisms. As consequence of these considerations the authors  propagate intravesical treatment of antibiotic therapy by which high enough concentrations can be build up immediately. So far so good.

Here are my comments/questions.

1. It is not clear throughout the manuscript, whether the authors in this manuscript only discuss treatment of uncomplicated cystitis or cystitis of any case, also including different kinds of complicated cystitis. To my opinion this concept of intravesical application only makes clinical sense if propagated for treatment of uncomplicated cystitis with single dose therapy. It is hard to imagine that intravesical treatment will be repeated several time if needed, e.g. in patients with complicated cystitis without indwelling catheter or on intermittent catheterization. At least in the text it is not always clear, what kind of cystitis in meant in a specific paragraph: treatment of uncomplicated or complicated cystitis; OR: prevention of recurrences in recurrent uncomplicated or complicated cystitis. If the patients with an indwelling catheter or on intermittent catheterization because of neurogenic bladder disturbances, of course repeated vesical instillation is acceptable.

2. For treatment of an acute episode of uncomplicated cystitis according to the EAU guidelines only the following oral antibiotics are recommended as first line – and in practice are only used for treatment of UTI at least in Germany-: fosfomycin trometamol, nitrofurantoin (treatment and prophylaxis) and pivmecillinam and in addition according to the German AWMF guidelines: nitroxoline. According to the Antibiotic Resitance Surveillance of the German Robert Koch Institut of 2022 the resistance of all E. coli against  Fosfomycin was 1.6% and against nitrofurantoin was 0.9%. The resistance against other anitibiotics which are not first-line recommended for treatment of uncomplicated cystits, but also used for many other infections are already faily high: Co-Amoxiclav 32.5%, Co-Trimoxazole 19.8%, Ciprofloxacin 13.8%. From these data it does not seem to be obvious, that resistance of nitrofurantoin is triggered by oral treatment  of cystitis or prophylaxis for recurrent cystitis. Nevertheless – as discussed by the authors – especially for treatment of uncomplicated cystitis and prophylaxis of recurrent uncomplicated cystitis non-antimicrobial methods should be investigated and propagated. Although I could imagine a following study for an acute episode of uncomplicated cystitis: single dose intravesical instillation of gentamycin at the moment after catheter urine is taken for microbiology vs oral nitrofuarntoin or single does of fosfomycin trometamol. At least from the abstracts of the paper cited I have not seen such a study.

3. Most studies cited with intravesical administration of gentamycin refer to complicated cystitis e.g. in patients with neurogenic bladder e.g. on intermittent catheterization or for prevention after urogynecological operation. This procedure makes sense and should be propagated.

4. Of course, I agree that intravesical treatment of lower UTI should be discussed seriously an the basis of PK/PD considerations. However, for clinical use and recommendation we need to define the clinical situation and the underlying complicating factors, whether in a specific situation such an approach may be helpful or even superior to oral treatment. As far as I can see in the references 144-150, studies using intravesical gentamicin are discussed with very specific underlying conditions.

Otherwise I have no further comments.

Author Response

-----As consequence of these considerations the authors  propagate intravesical treatment of antibiotic therapy by which high enough concentrations can be build up immediately. So far so good.

Response: We highly appreciate the reviewer’s remark, which encapsulates the central premise of the review in nutshell.

Here are my comments/questions.

  1. It is not clear throughout the manuscript, whether the authors in this manuscript only discuss treatment of uncomplicated cystitis or cystitis of any case, also including different kinds of complicated cystitis. To my opinion this concept of intravesical application only makes clinical sense if propagated for treatment of uncomplicated cystitis with single dose therapy. It is hard to imagine that intravesical treatment will be repeated several time if needed, e.g. in patients with complicated cystitis without indwelling catheter or on intermittent catheterization. At least in the text it is not always clear, what kind of cystitis in meant in a specific paragraph: treatment of uncomplicated or complicated cystitis; OR: prevention of recurrences in recurrent uncomplicated or complicated cystitis. If the patients with an indwelling catheter or on intermittent catheterization because of neurogenic bladder disturbances, of course repeated vesical instillation is acceptable.

Response: We thank the reviewer for the constructive comment, and we would like to clarify that as the title of the manuscript also states that manuscript specifically elaborates on the strength, weakness, opportunities, and threats (SWOT) of oral antimicrobial treatment (OAT), primarily prescribed for uncomplicated cystitis. The weakness and threats of OAT stemming from poor bioavailability and renal insufficiency are thrown into sharp relief by the immediate delivery of antimicrobials via intravesical administration for recurrent infection of uncomplicated cystitis. Since intravesical treatment is yet to receive regulatory approval for uncomplicated cystitis, it would be premature to comment on the appropriate regimen and frequency of intravesical administration for efficacy against recurrent infections. Moreover, instead of following the old concept of “one dose fits all”, modern era of personalized medicine is epitomized by individualization of antibiotics as well as dose by susceptibility of bacteria and individual patient condition with or without indwelling catheter or recurrent infection, as also noted by reviewer. However, we do not envision, intravesical administration of antimicrobials for complicated cystitis which is marked by infection ascent and urosepsis but intravesical administration can certainly circumvent the weaknesses of OAT in the management of uncomplicated cystitis with immediate accomplishment of MIC to counter AMR development.

  1. For treatment of an acute episode of uncomplicated cystitis according to the EAU guidelines only the following oral antibiotics are recommended as first line – and in practice are only used for treatment of UTI at least in Germany-: fosfomycin trometamol, nitrofurantoin (treatment and prophylaxis) and pivmecillinam and in addition according to the German AWMF guidelines: nitroxoline. According to the Antibiotic Resitance Surveillance of the German Robert Koch Institut of 2022 the resistance of all E. coli against  Fosfomycin was 1.6% and against nitrofurantoin was 0.9%. The resistance against other anitibiotics which are not first-line recommended for treatment of uncomplicated cystits, but also used for many other infections are already faily high: Co-Amoxiclav 32.5%, Co-Trimoxazole 19.8%, Ciprofloxacin 13.8%. From these data it does not seem to be obvious, that resistance of nitrofurantoin is triggered by oral treatment  of cystitis or prophylaxis for recurrent cystitis. Nevertheless – as discussed by the authors – especially for treatment of uncomplicated cystitis and prophylaxis of recurrent uncomplicated cystitis non-antimicrobial methods should be investigated and propagated. Although I could imagine a following study for an acute episode of uncomplicated cystitis: single dose intravesical instillation of gentamycin at the moment after catheter urine is taken for microbiology vs oral nitrofuarntoin or single does of fosfomycin trometamol. At least from the abstracts of the paper cited I have not seen such a study.

Response: We thank the reviewer for insightful remark and sharing data on the low incidence of AMR from OAT of cystitis in Germany, which can be dramatically different from incidence rates of AMR with first line antibiotics in Saudi Arabia (PMID: 38476810). Role of geographical and cultural differences in antibiotic prescribing behavior can certainly impact AMR incidences. As we stated in response to the previous statement, our goal is not to advocate intravesical antibiotics for uncomplicated cystitis with this review because intravesical treatment and regimen are yet to be approved by regulatory agencies. Nevertheless, clinical outcomes of OAT and intravesical administration of antimicrobials can support the logical argument on SWOT and root cause analysis of AMR developed with OAT in uncomplicated cystitis.

  1. Most studies cited with intravesical administration of gentamycin refer to complicated cystitis e.g. in patients with neurogenic bladder e.g. on intermittent catheterization or for prevention after urogynecological operation. This procedure makes sense and should be propagated.

Response: We are very heartened by the insightful comment. We completely agree with reviewer’s point of view for complicated cystitis and so wish to share information with the reviewers about developments underway to enable regulatory approval of such treatments, albeit this is somewhat of a tangential digression from the analyses presented in this article.  Although regulatory agencies may not prioritize the development of intravesical products for cystitis, a growing body of clinical evidence on off-label intravesical use of antibiotics, most notably aminoglycosides, for people with catheters, catheter associated cystitis prophylaxis as well as treatments of uncomplicated or complicated cystitis may compel regulatory agencies to grant approval to intravesical administration of antibiotics or non-antibiotic antimicrobials as potential treatment for recurrent infections and prophylaxis for uncomplicated cystitis. There has been a development in catheter technology, although not yet available, for instillation of a therapeutic through a sterile channel running from a syringe port, intramurally through the length of a catheter that will enable delivery of a sterile and accurately dosed treatment into the bladder, immediately following urinary drainage and closure of the drainage lumen (to prevent instillation efflux through the drainage channel). However, until such technology is licensed by the regulators, we believe it would be imprudent to raise this matter in this article. Nevertheless, we believe it is crucial for the science presented to be understood now by clinicians involved in treating cystitis as direct targeting of inoculated bacteria with a catheter’s insertion, with a mutation prevention concentration (MPC) of drug before they have any chance to develop their defenses is a scientifically sound strategy.

  1. Of course, I agree that intravesical treatment of lower UTI should be discussed seriously an the basis of PK/PD considerations. However, for clinical use and recommendation we need to define the clinical situation and the underlying complicating factors, whether in a specific situation such an approach may be helpful or even superior to oral treatment. As far as I can see in the references 144-150, studies using intravesical gentamicin are discussed with very specific underlying conditions.

Response: We thank the reviewer for the insightful comment, and we discussed intravesical administration of microbials to highlight the PK/PD limitations of OAT in treatment of uncomplicated cystitis. As we stated in response to earlier comments, in absence of regulatory approval, it would be premature on our part to make recommendation for intravesical treatment beyond the conditions in cited studies on intravesical gentamicin.

Round 2

Reviewer 1 Report (New Reviewer)

Comments and Suggestions for Authors

The authors have significantly improved the article. However, the only suggestion would be to add a brief section in the discussion describing Polymer Ligating Clips in Urologic Surgery. The discussions could be related to the evolution of materials in urology, resulting in enhanced efficacy in treating pathologies, especially those related to the bladder and consequently reducing risks of infections, cystitis, hemorrhages and severe conditions.

Comments on the Quality of English Language

Minor editing of English language required.

Author Response

We thank the reviewer for constructive comment, and we have now added a new paragraph highlighted in yellow from 418 to 436 on polymer modifications to retard biofilm forming uropathogens which cause CAUTI. While a discussion on infection belongs to this manuscript, we believe discussion on polymer ligating clips in urologic surgery is beyond the scope of this narrative. We have also corrected English language throughout the manuscript.

This manuscript is a resubmission of an earlier submission. The following is a list of the peer review reports and author responses from that submission.

Round 1

Reviewer 1 Report

Comments and Suggestions for Authors

This review entitled as “SWOT and Root Cause Analysis of Antimicrobial Resistance to Oral Antimicrobial Treatment of Cystitis” explores the impact of oral antimicrobial therapy (OAT) on antimicrobial resistance (AMR) in cystitis. It discusses the physiological factors influencing OAT efficacy, advocates for a distinct approach to cystitis treatment, and suggests alternative strategies, emphasizing the need for judicious antibiotic use to combat evolving AMR challenges. However, despite acknowledging the successes of OAT, there are some aspects that could be considered for improvement:

1.      Abstract

a.       The conclusion seems abrupt and doesn't effectively summarize the main findings or emphasize the significance of the research. A more impactful conclusion could leave a lasting impression on the reader.

2.      Introduction

a.       The introduction primarily outlines facts but lacks critical evaluation or analysis of the current state of knowledge in the field. Engage more deeply with existing literature, highlighting gaps and areas of controversy.

b.      The mention of Figure 1 and its relevance to the threats faced by uropathogens is promising. However, for a more comprehensive understanding, consider briefly describing or referencing the figure in the text.

c.       While mentioning the SWOT analysis and root cause analysis (RCA) is beneficial, consider briefly elaborating on what aspects these analyses will cover. This could provide readers with a roadmap for what to expect in the subsequent sections.

3.      Antimicrobial resistance (AMR)

a.       The statement that "efforts to contain AMR are capturing the attention of healthcare providers now" (line 49) is somewhat vague and overgeneralized. Provide specific examples or evidence to support this claim.

b.      The mention of non-compliance with guidelines and antimicrobial stewardship (AMS) as major contributors to AMR (lines 65-66) could be expanded. Provide examples or elaborate on how these issues specifically lead to increased resistance.

c.       The prospective study on 1758 women is mentioned, but the context and key findings are not thoroughly explained. Provide more information on the study design and its relevance to the discussion.

4.      In the section 3, The Urinary Bladder, the narrative lacks a seamless connection between the bladder's role in homeostasis and subsequent discussions on urothelium and neutrophils. Strengthen the logical progression of ideas to create a more cohesive and understandable flow.

5.      In the section 4, Cystitis Etiology and Treatment, The mention of critical elements of cystitis pathogenesis identified in mouse models not being reproducible in the pig model requires further elaboration. Explain why the pig model is considered more reflective of human pathogenesis and its implications for research.

6.      Root cause analysis of OAT contribution to AMR

a.       The statement regarding the regulatory approval of intravesical antimicrobial therapy (IAT) lacks specificity. Clarify the current status of regulatory approval and any ongoing developments in this area.

b.      The statement "Cystitis treatment failures and AMR result from a bladder milieu that allows adaptable microbes to evade, escape, or expel antimicrobials" lacks specificity and clarity regarding the mechanisms involved. Offer specific instances or elucidate the mechanisms that demonstrate the bladder environment's role in contributing to treatment failures and the development of antimicrobial resistance (AMR).

7.      Discussion

a.       The discussion lacks specific examples or case studies to support the claim that current OAT prescriptions contribute significantly to AMR. Providing concrete instances or referring to relevant studies would enhance the credibility of the argument.

b.      The terms "intracavity" and "topical" used in relation to IAT for cystitis (Fig.4) need clarification. The distinction between these terms is not well-defined, leading to potential confusion for readers

c.       While the discussion briefly mentions alternatives such as intravesical instillation of innocuous microbes, antimicrobial peptides, and bacteriophage treatment, there is insufficient analysis or comparison of these alternatives. A more in-depth exploration of the advantages and disadvantages of each approach would strengthen the argument.

8.  

Author Response

This review entitled as “SWOT and Root Cause Analysis of Antimicrobial Resistance to Oral Antimicrobial Treatment of Cystitis” explores the impact of oral antimicrobial therapy (OAT) on antimicrobial resistance (AMR) in cystitis. It discusses the physiological factors influencing OAT efficacy, advocates for a distinct approach to cystitis treatment, and suggests alternative strategies, emphasizing the need for judicious antibiotic use to combat evolving AMR challenges. However, despite acknowledging the successes of OAT, there are some aspects that could be considered for improvement:

Response: We thank the reviewer for highlighting the positive attributes and areas for improvement in the manuscript. We have addressed those areas in revised manuscript.

  1. Abstract
  2. The conclusion seems abrupt and doesn't effectively summarize the main findings or emphasize the significance of the research. A more impactful conclusion could leave a lasting impression on the reader.

Response: We thank the reviewer for the constructive comment, and the abstract is now rewritten to generate an impactful conclusion.  

  1. Introduction
  2. The introduction primarily outlines facts but lacks critical evaluation or analysis of the current state of knowledge in the field. Engage more deeply with existing literature, highlighting gaps and areas of controversy.

Response: We thank the reviewer for the comment, and the introduction is now extensively revised. The review analyzes the complex problem of AMR encountered by antimicrobial drug in combating uropathogen localized in bladder urothelium. Instead of focusing on the different attributes of adaptable uropathogen driving AMR, the review is focused on oral drugs, specifically their time dependent arrival and interaction (pharmacokinetics) that delays the arrival of MIC in urine and weaken their potency in combating uropathogen. There is insufficient literature on the clinical significance of the delayed arrival of orally absorbed drug molecules to bladder leads to uropathogens getting exposed to below MIC concentration in urine and descendant colonies bestowed with AMR gain an upper hand in combat with the antimicrobial. As opposed to oral administration, intravesical administration of antimicrobial avoids the delay in reaching urinary MIC to kill uropathogen. Therefore, the central theme of this review is to elucidate the significance of the delayed MIC in urine.

  1. The mention of Figure 1 and its relevance to the threats faced by uropathogens is promising. However, for a more comprehensive understanding, consider briefly describing or referencing the figure in the text.

Response: We thank the reviewer for the comment and figure 1 is now described in the legend, introduction and in text following the figure.

  1. While mentioning the SWOT analysis and root cause analysis (RCA) is beneficial, consider briefly elaborating on what aspects these analyses will cover. This could provide readers with a roadmap for what to expect in the subsequent sections.

Response: We thank the reviewer for the comment, and we have now provided a roadmap for the subsequent sections in the introduction with explanation of SWOT and RCA.

  1. Antimicrobial resistance (AMR)
  2. The statement that "efforts to contain AMR are capturing the attention of healthcare providers now" (line 49) is somewhat vague and overgeneralized. Provide specific examples or evidence to support this claim.

Response: We thank the reviewer for the comment, and we have now connected the problem of AMR with the global initiative of AMS in the revised manuscript.

  1. The mention of non-compliance with guidelines and antimicrobial stewardship (AMS) as major contributors to AMR (lines 65-66) could be expanded. Provide examples or elaborate on how these issues specifically lead to increased resistance.

Response: We thank the reviewer for the comment, and we have now elaborated on that aspect by citing studies (reference 30, 35,37 and 38 of the manuscript) documenting antibiotic overuse leading to AMR which triggered the practice of AMS.

  1. The prospective study on 1758 women is mentioned, but the context and key findings are not thoroughly explained. Provide more information on the study design and its relevance to the discussion.

Response: We thank the reviewer for the comment and we have now extensively revised the discussion of strengths and limitations of study in the revised manuscript.

  1. In the section 3, The Urinary Bladder, the narrative lacks a seamless connection between the bladder's role in homeostasis and subsequent discussions on urothelium and neutrophils. Strengthen the logical progression of ideas to create a more cohesive and understandable flow.

Response: We thank the reviewer for the comment. We have now added text to connect the two sections in the revised manuscript

  1. In the section 4, Cystitis Etiology and Treatment, The mention of critical elements of cystitis pathogenesis identified in mouse models not being reproducible in the pig model requires further elaboration. Explain why the pig model is considered more reflective of human pathogenesis and its implications for research.

Response: We thank the reviewer for the critical analysis and have now added details of colony forming units (CFU) and bladder wall thickness, evolutionary distance of humans to mice relative to pigs in the revision. While small size of mice enables their widespread use for studying cystitis pathogenesis, thinner bladder wall of mice may be altering the dynamics of infection as embodied by intracellular bacterial colonies following inoculation of UPEC at thousand-fold higher bacterial loads than bacterial loads per mL of urine in humans. Importantly, the observation of intracellular bacterial colonies in mouse bladder is yet to be authenticated in humans for their validity and clinical predictability. However, it is worth considering that pig bladder wall is similar in size to humans and bacterial inoculation at bacterial loads comparable to humans does not lead to intracellular bacterial colonies. However, pig research has not matured yet to merit superiority and judging animal models of UTI is beyond the scope of this article as these profound questions have economic considerations as well.  Pig models can potentially serve as a bridge or gatekeeper of observations made in mice.

  1. Root cause analysis of OAT contribution to AMR
  2. The statement regarding the regulatory approval of intravesical antimicrobial therapy (IAT) lacks specificity. Clarify the current status of regulatory approval and any ongoing developments in this area.

Response: We thank the reviewer and as stated in the review IAT is yet to receive regulatory approval and we emphatically made the negative statement on approval of IAT.

  1. The statement "Cystitis treatment failures and AMR result from a bladder milieu that allows adaptable microbes to evade, escape, or expel antimicrobials" lacks specificity and clarity regarding the mechanisms involved. Offer specific instances or elucidate the mechanisms that demonstrate the bladder environment's role in contributing to treatment failures and the development of antimicrobial resistance (AMR).

Response: We thank the reviewer for the comment and the mechanisms of AMR now described in detail in root cause analysis section 4.2 and again in discussion. We have added reference for each mechanism of AMR in root cause analysis.  

  1. Discussion
  2. The discussion lacks specific examples or case studies to support the claim that current OAT prescriptions contribute significantly to AMR. Providing concrete instances or referring to relevant studies would enhance the credibility of the argument.

Response: We thank the reviewer for the comment and we have added concrete instances and relevant studies (reference 30, 35,37 and 38 of the manuscript) in the section 1 on AMR and again in the discussion for enhanced credibility.

  1. The terms "intracavity" and "topical" used in relation to IAT for cystitis (Fig.4) need clarification. The distinction between these terms is not well-defined, leading to potential confusion for readers.

Response: We thank the reviewer for the comment, we have now deleted those terms to avoid confusion.

  1. While the discussion briefly mentions alternatives such as intravesical instillation of innocuous microbes, antimicrobial peptides, and bacteriophage treatment, there is insufficient analysis or comparison of these alternatives. A more in-depth exploration of the advantages and disadvantages of each approach would strengthen the argument.

Response: We thank the reviewer for the comment and these approaches are still in exploration phase and it is still too early to make a comparative determination. No study has been done on the comparative effectiveness of so many non-antibiotic approaches tested so far.

Reviewer 2 Report

Comments and Suggestions for Authors

This paper is overreaching and addresses too many broad topics, and as a result the main subject of the paper is unclear to the reviewers. The study attempts to focus on the correlation between resistance and oral antibiotics, but never actually provided any good evidence or references to suggest any such correlation. The SWOT and Root Cause analysis are also not well established here.  Line 28 – Needs a reference

Line 30 – this reference only talks about H. pylori, can it be applied in general to all microbes?

Line 35 – reference #7 is incorrect

Line 36 – this reference is not really relevant to this comment

Line 39 – not sure if “intrinsic deficiencies” is an appropriate term

Line 44 – “modest success” maybe too diminishing and oversimplifying the effect of AMS? Also needs references for this 

Line 49 – what action? This number is a little too specific – maybe just “nearly 300,000”?

Line 52 – 3.8 billions years again? Not sure why this number keeps coming up

Line 63 – First line is nitrofurantoin, not Bactrim

Line 75/76 – needs a reference for this

Line 79 – from the referenced article: Previous IV or oral antimicrobial use in the past 90-days and history of a multi-drug resistant pathogen were associated with FQ-resistant E. coli (odds ratio [OR] 2.68, 95% confidence interval [CI]: 2.04-3.51, and OR 6.93, 95% CI: 4.95-9.70, respectively)

2.2 AMR and Pregnancy – this section is confusing and not accurate, UTI in pregnant patient is indicated – not contraindicated - for treatment, and not all oral antibiotics are teratogenic. The referenced study was conducted in India, and for their limitations they stated “the exact data about antibiotic consumption in the recent past could not be obtained from the patients”. It would be inappropriate for the authors to conclude here that significantly higher incidence of AMR is attributed to OAT used by non-pregnant women.

Line 104 – nocturia is not an established symptom of cystitis, it is a symptom of interstitial cystitis. Do the authors have a reference for this?

Line 125 – Isn’t this true for humans, not only in rodent models

Line 140/141 – this sentence is confusing and does not match the reference

Line 147 – defined by which guideline?

Line 151/152 – this sentence does not match the reference

Line 155 – how is multiple sclerosis relevant to this topic?

Line 158-164 - This whole paragraph is confusing, why are we comparing mouse models to pigs and then human? If pig models are better than mice, then the authors should better support this claim with references

Line 173-175 – this sentence does not make sense – would re-write

Line 180/181 – does this not apply to parenteral antibiotics as well? What is the variability in clinical outcomes?

4.1 SWOT of OAT for UTI: the whole section again is confusing, and there are a lot of unreferenced and incorrect information. SWOT analysis was not appropriately conducted. One strength and weakness were discussed as opposed to listing all major components, opportunities and threat as described by this approach were not appropriately used here.

Line 304 – was AMS already explained in a prior paragraph?

5 Root cause analysis: Root cause analysis needs to describe and rule out other causes in a step-wise manner in order to arrive at the root cause, however the author’s identification of the root cause did not discuss other possible causes and rule them out for the reader to understand

this paragraph talks about data from various geographical location, these different countries could have very different AMR patterns and antibiotic usage. Again “modest success” is not sufficiently explained here with enough references. The authors talk about intravesical delivery of gentamicin and how it complies with the 5Ds – they should instead look into why intravesical antibiotics are not recommended by guidelines for simple cystitis as first line and why it’s outdated

6 Discussion: this section suffers similar problems as the other sections, would edit accordingly after reworking the other paragraphs. Patient with spinal cord injury or with chronic catheter use are not considered simple cystitis patients. Clinical data of the use of oral antibiotics for cystitis treatment is very robust, and we suggest that the authors do more research on the topic before reworking this paper. 

Author Response

This paper is overreaching and addresses too many broad topics, and as a result the main subject of the paper is unclear to the reviewers. The study attempts to focus on the correlation between resistance and oral antibiotics, but never actually provided any good evidence or references to suggest any such correlation. The SWOT and Root Cause analysis are also not well established here.  

Response: We thank the reviewer for the critical appraisal of this interdisciplinary look at AMR associated with OAT for cystitis. There have been plethora of review articles discussing the role of genetic and phenotypic attributes of uropathogen in driving AMR. Here, we choose to focus on the antimicrobial drug and its time dependent interaction (pharmacokinetics) with the human body, especially intravesical space in AMR development. Instead of a piecemeal approach, we performed a comprehensive analysis of factors involved in the delayed arrival of MIC in urine. There is insufficient literature on the clinical significance of the delayed arrival of orally absorbed drug molecules to bladder which leads to uropathogens getting exposed to below MIC concentration in urine and descendant colonies bestowed with AMR gaining an upper hand in combat with the antimicrobial. The central theme of this review is to elucidate the significance of the delayed MIC in urine through a fact-based discussion on determinants of AMR in root cause analysis and SWOT of OAT. The weight of published evidence implicates a strong correlation between resistance and oral antibiotic use for cystitis and we have increased the reference list by 40%.

Line 28 – Needs a reference

Response: We thank the reviewer for careful review and we have added the reference, new reference 1.

Line 30 – this reference only talks about H. pylori, can it be applied in general to all microbes?

Response: We thank the reviewer for the comment and the citation of an article on H. pylori was predicated on microbes capable of surviving in acidic pH sharing survival mechanisms with uropathogens capable of surviving hyperosmotic urine.

Line 35 – reference #7 is incorrect

Response: We respectfully disagree as the reference in question is correct because it refers to the variability in pH and osmolality of urine stemming from the variable reabsorption of urine constituents by bladder. The urine osmolality is also relevant to the previous comment.  We have clarified the statement by adding the adjective “stored”.

Line 36 – this reference is not really relevant to this comment

Response: We respectfully disagree with the reviewer’s opinion on this matter as new reference 10 and 11 refers to the microbiome existing in bladder of healthy individuals known to counter the invasion of uropathogens and therefore, we respectfully submit that references are relevant to the statement in manuscript.

Line 39 – not sure if “intrinsic deficiencies” is an appropriate term

Response: We thank the reviewer for comment and in the revised manuscript, the term intrinsic deficiency is replaced by weaknesses, to link up with W of SWOT.

Line 44 – “modest success” maybe too diminishing and oversimplifying the effect of AMS? Also needs references for this 

Response: We thank the reviewer for comment and in the revised manuscript, we have cited the references on modest success of AMS.

Line 49 – what action? This number is a little too specific – maybe just “nearly 300,000”?

Response: We thank the reviewer for the comment. The action refers to the worldwide effort of all the stakeholders of health, which is captured in 249866 articles up from 249,067 in pubmed research performed in January 2024. We believe it would be premature to round up the number specified by reviewer, which would be reached in a year or so.

Line 52 – 3.8 billions years again? Not sure why this number keeps coming up

Response: We thank the reviewer for critical comment, the number is essential is drive home the point about survivability and adaptability of uropathogens and that number is now referenced by two published articles as asked by the reviewer in the first comment.

Line 63 – First line is nitrofurantoin, not Bactrim

Response: We thank the reviewer for the critical comment, and we have now revised the order of treatment

Line 75/76 – needs a reference for this

Response: We thank the reviewer for the critical comment and we have now added two references for this statement

Line 79 – from the referenced article: Previous IV or oral antimicrobial use in the past 90-days and history of a multi-drug resistant pathogen were associated with FQ-resistant E. coli (odds ratio [OR] 2.68, 95% confidence interval [CI]: 2.04-3.51, and OR 6.93, 95% CI: 4.95-9.70, respectively)

 Response: We thank the reviewer for the critical comment and we revised the statement in question by adding FQ-resistance.

2.2 AMR and Pregnancy – this section is confusing and not accurate, UTI in pregnant patient is indicated – not contraindicated - for treatment, and not all oral antibiotics are teratogenic. The referenced study was conducted in India, and for their limitations they stated “the exact data about antibiotic consumption in the recent past could not be obtained from the patients”. It would be inappropriate for the authors to conclude here that significantly higher incidence of AMR is attributed to OAT used by non-pregnant women.

 Response: We thank the reviewer for highlighting the confusion in the section which was necessitated by the 50% lifetime incidence of UTI and even higher incidence of pregnancy in women. The events of UTI and pregnancy are certainly not mutually exclusive as evident from the 8% incidence of UTI in pregnant women which is certainly lower than the incidence of UTI in non-pregnant women. Lower incidence of UTI and the contraindication of teratogenic antibiotics during pregnancy could be plausibly linked to the lower antibiotic use by pregnant women which can explain lower AMR reported pregnant women of the study. The observation of higher AMR with higher antibiotic use is not unique to India, it is a worldwide phenomenon even seen in countries with electronic health records, documenting every drug and procedure performed on patients (reference 30, 35,37 and 38 of the manuscript). We have revised the statements accordingly.

Line 104 – nocturia is not an established symptom of cystitis, it is a symptom of interstitial cystitis. Do the authors have a reference for this?

 Response: We thank the reviewer for the critical comment and we have now removed the term nocturia from the discussion. Yes, nocturia is certainly a symptom of interstitial cystitis, but that is not relevant to the discussion on infective cystitis.

Line 125 – Isn’t this true for humans, not only in rodent models

 Response: We would like to clarify that mechanisms illustrated in Fig.1 are true for humans as well as rodents. Only the phenomenon of intracellular bacterial colonies is noted in rodents and not in pigs or human studies.

Line 140/141 – this sentence is confusing and does not match the reference

Response: We thank the reviewer for the critical comment and we have now revised the sentence to avoid confusion.

 Line 147 – defined by which guideline?

Response: We thank the reviewer for the critical comment. The symptoms of bacterial cystitis do not change across guidelines and same is true for the guidelines on the symptom based diagnosis of interstitial cystitis. The statements are supported by reference.

Line 151/152 – this sentence does not match the reference

Response: We thank the reviewer for the critical comment and we have now corrected the reference in question.

Line 155 – how is multiple sclerosis relevant to this topic?

Response: We thank the reviewer for the critical comment and cystitis is a complication associated with neurogenic bladder of multiple sclerosis patients or spinal cord injured patients. Accordingly, multiple sclerosis is relevant to the topic of cystitis.

Line 158-164 - This whole paragraph is confusing, why are we comparing mouse models to pigs and then human? If pig models are better than mice, then the authors should better support this claim with references

Response: We thank the reviewer for giving us the opportunity to clarify the paragraph. It is important to compare mouse models to pigs and humans because mechanistic information on cystitis pathogenesis is not derived from humans but from mouse studies. However, cystitis in mice is dissimilar to humans in many aspects including the requirement of thousand-fold higher bacterial loads for provoking cystitis in mice and transurethral inoculation of UPEC at high bacterial loads in mice may lead to the formation of intracellular bacterial colonies which are not seen with the inoculation of lower bacterial loads in pig bladder comparable to bacterial load in urine of cystitis patients. No human study has reported location of intracellular bacterial colonies in autopsy specimens or bladder biopsy. Since the formation of intracellular bacterial colonies creates a reservoir of bacteria in urothelium, there is a concern that cystitis pathogenesis in mice may be dissimilar from humans.  We have elaborated on this matter in revised manuscript. Reviewer 1 also made a similar comment on this matter, and we are repeating here in this response that making a firm determination of animal models of UTI is beyond the scope of this article. Moreover, pig research has not matured yet to make a firm determination on superiority of pig models as the questions has economic considerations as well. Nevertheless, the authentication with the ground truths of human observations is essential for the validity and clinical predictability of observations made in animal models, which is yet to happen for the observation of intracellular bacterial colonies. In that context, pig bladder wall is similar in size to humans and pig model can potentially serve as a bridge or gatekeeper of observations made in mice.

Line 173-175 – this sentence does not make sense – would re-write

Response: We thank the reviewer for the suggestion and we have now re-written the sentence in revised manuscript.

 Line 180/181 – does this not apply to parenteral antibiotics as well? What is the variability in clinical outcomes?

Response: We thank the reviewer for the comment. As now clearly stated in the title and in introduction, this review is focused on the enteral route by Oral antimicrobial therapy (OAT). The bioavailability from parenteral route is completely different from oral route as also hinted in the introduction. Parenteral route is not encumbered by the variability stemming from the variable absorption of drug from the gut, first pass metabolism in liver prior to arrival in bladder. Accordingly, the delay and variability of delay for reaching MIC in urine is much shorter than OAT, which lessens the likelihood for AMR emergence compared to oral route. The self-medication of OAT also does not burden doctors or nurses as is the case with the parenteral administration of antimicrobials.  The variability in clinical outcomes is treatment success marked by clear urine, or treatment failures marked by cystitis progressing to pyelonephritis or urosepsis and recurrent infections.

4.1 SWOT of OAT for UTI: the whole section again is confusing, and there are a lot of unreferenced and incorrect information. SWOT analysis was not appropriately conducted. One strength and weakness were discussed as opposed to listing all major components, opportunities and threat as described by this approach were not appropriately used here.

Response: We thank the reviewer for the critical review of the manuscript and are pleased with the opportunity to clarify. We have now described SWOT in narrative and in bullets at the end of section as reviewer advised.  We discussed the SWOT in narrative form in earlier version and the discussion of SWOT extended to other sections as well.

 Line 304 – was AMS already explained in a prior paragraph?

Response: Yes, AMS acronym was expanded upon in preceding sentence and AMS is now explained together with AMR in the section following introduction.

5 Root cause analysis: Root cause analysis needs to describe and rule out other causes in a step-wise manner in order to arrive at the root cause, however the author’s identification of the root cause did not discuss other possible causes and rule them out for the reader to understand this paragraph talks about data from various geographical location, these different countries could have very different AMR patterns and antibiotic usage. Again “modest success” is not sufficiently explained here with enough references. The authors talk about intravesical delivery of gentamicin and how it complies with the 5Ds – they should instead look into why intravesical antibiotics are not recommended by guidelines for simple cystitis as first line and why it’s outdated.

Response: We thank the reviewer for the detailed comment and the section of root cause analysis is now more streamlined into a coherent discussion. While antibiotic usage differs across countries, the menace of AMR is universal. Therefore, we limited the scope of this manuscript to the pharmacokinetic dimensions with respect to OAT and IAT. The root cause we identified is the initial delay in urinary MIC and we have identified the contribution of different factors in compounding that cause. While OAT is the first line treatment for simple cystitis, multiple lines of evidence now support the superiority of IAT over OAT in the treatment of recurrent cystitis. Our review article now cites seven retrospective and perspective studies on intravesical gentamicin use within the last four years to make a fact-based statement on its current use for prophylaxis. The reasons for guidelines not recommending IAT as first line treatment for simple cystitis is not because of pharmacological deficiencies in treatment but economics and availability of IAT compared to OAT. We listed five references on modest success of AMS in high income countries and we have raised that number in the revised manuscript.  

6 Discussion: this section suffers similar problems as the other sections, would edit accordingly after reworking the other paragraphs. Patient with spinal cord injury or with chronic catheter use are not considered simple cystitis patients. Clinical data of the use of oral antibiotics for cystitis treatment is very robust, and we suggest that the authors do more research on the topic before reworking this paper. 

Response: We thank the reviewer for the comment and would like to respectfully submit that the clinical co-authors have a combined clinical experience of over 50 years in prescribing OAT for cystitis in spinal cord injured, multiple sclerosis and aged patients.  Furthermore, the characterization of spinal cord injured patients and patients with chronic catheter as not having simple cystitis is contested by extensive recruitment of patients from this patient pool for studies testing non-antibiotic approaches (implantation of innocuous microbes, phages, glycoaminoglycans and antimicrobial peptides) as prophylaxis and the treatment of cystitis. High prevalence of asymptomatic bacteriuria in this patient population warrants non-antibiotic approaches. As discussed in the manuscript, no one disputes that the clinical data on use of OAT for cystitis is robust but the evidence for the prevalence of AMR with the use of OAT is also getting equally robust.

Round 2

Reviewer 1 Report

Comments and Suggestions for Authors

The manuscript has improved since the comments were addressed.

Author Response

We thank the reviewer for critical review of the manuscript.

Reviewer 2 Report

Comments and Suggestions for Authors

The manuscript still has incorrect references and misinformation, such as the recommended treatment for cystitis in pregnancy. The authors also failed to improve the SWOT and Root cause analysis, since both are still missing significant information regarding the strengths and weaknesses of oral antibiotics. 

Author Response

Comment: The manuscript still has incorrect references and misinformation, such as the recommended treatment for cystitis in pregnancy. The authors also failed to improve the SWOT and Root cause analysis, since both are still missing significant information regarding the strengths and weaknesses of oral antibiotics. 

Response: We thank the reviewer for the critical analysis of our review, but we respectfully submit that the section 2.2 on pregnancy and AMR is well referenced and does not merit the remark of “incorrect references and misinformation”. However, on further reflection, we concede that the cystitis treatment of pregnant women is not the main point of the review and instead of contesting that issue any further, we have elected to remove the 2.2 section on AMR and pregnancy from the revised manuscript.

We have now added 40 more references than the references in the original manuscript and those new references merited extensive changes in the revised manuscript in response to specific comments to the original version of manuscript. The revised manuscript also discusses a new antibiotic, Gepotidacin for nitrofurantoin cystitis published last week in Lancet (references 25,26). As reviewer 1 also noted, revised manuscript is much stronger than the original version. Therefore, the above stated comment on “missing significant information” for SWOT and root cause analysis is vague and tough to reconcile with the extensive changes in the revised manuscript. It would have been helpful if reviewer would have pointed out what is specifically missing in the revised version as he did with the review of original version of this manuscript.

The word “primary” in paragraph 1 of section 4.1 under SWOT clearly outlines the constraints we imposed on our SWOT analysis. We do not claim our manuscript to be exhaustive of all aspects and minutiae but elected to include only the primary strengths and weaknesses in order to steer focus towards the essence of our root cause analysis.  We have been careful to present this article as a critical analysis based on established science and deductive analysis. However, underpinning the story is our concern that oral antimicrobial treatments predispose to failures and downstream AMR, which have hitherto gone unnoticed. Clinicians are blamed for overusing treatments rather than anyone having investigated the route of administration because efficacy of the drug is determined by the kinetics of drug concentration achieved at the site of infection.

Round 3

Reviewer 2 Report

Comments and Suggestions for Authors

Abstract

Line 11-12 first sentence does not flow correctly, it is not 25% out of 150 million cases of UTI, but out of outpatient oral antibiotics.

Introduction

Line 48-51 confusing to read.

Line 54 this reference is for oral Fosfomycin, do you have a reference that talks about why parenteral antibiotics would also have a delay in urinary MIC?

Line 60 it’s good that you included gepotidacin as a novel oral antibiotic for UTI, but nowhere does these clinical trials discuss delay in urinary MIC after administration, where are you inferring this information?

Line 61 what is an asymmetric advantage?

Antimicrobial resistance

Line 77-79 this reference does not match the sentence, what is a clinical governance?

Line 86 Postulate is used incorrectly here.

Line 90-92 needs a reference.

Line 96-100 sentence should be reworded, it’s confusing to include both US and UK data in one sentence.

Line 109-115 again this key study data is not correctly represented and this is another example of error in data evaluation.

The urinary bladder

3.1 neutrophils – while this section is interesting, we failed to see the relevance of this section to the rest of the paper. Would just keep the first paragraph of this big section.

Line 170-173 this statement doesn’t make sense, decline in chemokines after treating cystitis with oral antibiotics should indicate good resolution of infection, but are you saying this is a negative thing because of the lack of neutrophils?

Cystitis Etiology and Treatment

Line 183-186 why discuss urothelial carcinoma? We see no relevance to the article.

Line 193-194 confusing.

Line 197 no reference.

Line 199-201 this needs a reference, imaging is usually not needed when diagnosing complicated vs uncomplicated UTI.

Line 201-204 this sentence is confusing, why is the need for translational research important here? Are you stating that since mouse models cannot accurately predict human pathophysiology, we need more studies in pig bladder? But why is translational research important in the first place? There are many in vivo human studies of oral antibiotics and UTI.

Line 224-227 can’t this inter-individual variability be seen in other routes of administration?

Line 257 this reference is for Ertapenem, which is a parenteral antibiotic, the authors cannot infer threats to oral antibiotic from this reference.

Line 259 define infection ascent.

Line 261 overuse of antibiotics in general? Or just oral antibiotics. This reference refers to antibiotics in general.

Line 265 weakness – this sentence is made up, and the references listed here do not support this statement.

Line 268 opportunities - we do have predictable oral bioavailability for most oral antibiotics, where is the reference to this unpredictability? The reference listed here does not discuss this unpredictability. This statement is mostly made-up as well, are you saying oral antibiotics with 100% bioavailability will not encounter AMR?

Line 277-288 this section is confusing to read, and the authors should take a closer look at reference 20 to summarize their PK/PD findings for nitrofurantoin, and why it’s on indicated for uncomplicated cystitis and not systemic infections. Reference 101 talks about lidocaine, which is not relevant to this section. Reference 102 is a murine study on pregnant mice, since you already discussed previously that murine models are not enough to predict human data, why include it here?

Line 293 “errors in treatment of cystitis” not sure what this means. Are you saying we are not effectively treating cystitis? Where is the reference for this?

Line 205 “toxicity and lower therapeutic index” what does this mean?

Line 304 this statement is disingenuous and misleading since the reference only demonstrated that Fosfomycin and polymyxin needs this much higher of a concentration.

Line 307 “military doctrine”?

Figure 3 – what is the unit for Time here? What do you mean “mirage” and “unknowledge and indeterminable” real world data points? Is this graph accurate at all?

Line 330 “window of opportunity” do you have reference to back up this claim of development of resistance after first dose? Reference 107-108 discuss resistance but not in this context.

Line 335-336, the reference listed here does not match the statement.

Line 363 why talk about dog models? Did the authors focus on pig bladder models earlier? This data is also contradictory to the core hypothesis of this paper, since in some of the dogs they found faster Tmax in urine than serum concentrations.

Discussion

Line 382-389 not sure why this paragraph is needed here.

Line 391 “tortuous”?

Line 393-395 which experts? This reference is from 1979 and maybe not as relevant to modern practice. Can you find a more recent reference to this expert opinion on healthy voiding?

Line 397 the authors have not defined ASB yet, and asymptomatic bacteriuria is an important stewardship topic, if the authors want to focus on this instead, would recommend reading guidelines and discuss why it is not recommended to treat ASB.

Line 399-401 would reword this sentence, again data from other countries should not be used to infer US data.

Line 403-405 this was basically the same sentence from line 96-100.

Line 421-469 this was the same Root Cause analysis from before, but just moved to a different section. This section also remained mostly unchanged. This also didn’t fully address our initial edits. Again, are you saying intravesical antibiotics are better and should be given to all patients with UTIs, instead of oral antibiotics?

Line 479-482 most guidelines already identify cystitis as its own infection and should be treated differently from other complicated UTIs.

Line 508 where is the actual reference for this? Dequalinium chloride is not routinely used and is mostly indicated for throat infections, how is this relevant to UTI?

Line 515-518 not sure what you are trying to say here, guidelines are constantly changing based on new clinical data and new antibiotics.

Author Response

We thank the reviewer for constructive criticism, and we have now revised the manuscript with changes highlighted in yellow.

Line 11-12 first sentence does not flow correctly, it is not 25% out of 150 million cases of UTI, but out of outpatient oral antibiotics.

 Response: We thank the reviewer for this suggestion, and we have now rectified the same in revision.

Introduction

Line 48-51 confusing to read.

Response: We are thankful for the opportunity to revise, and we have tried our best in clarifying the significance of plasma and urine levels of antimicrobials in treating bacteremia and cystitis, respectively. However, we could not help noticing that any discussion on the nuances of pharmacokinetics that are relevant for antimicrobial treatment of cystitis, be it in SWOT or root cause analysis of AMR has invariably earned the label “confusing to read” from the reviewer. If we posit that the ignorance of pharmacokinetics among prescribers is central to the problem of AMR in the treatment of uncomplicated cystitis, then we cannot begin to understand the problem, let alone solve till prescribers vehemently resist any discussion on pharmacokinetics in relation to AMR. As Einstein reportedly said, “We can't solve problems by using the same kind of thinking we used when we created them.”

Line 54 this reference is for oral Fosfomycin, do you have a reference that talks about why parenteral antibiotics would also have a delay in urinary MIC?

Response: We were bit bewildered by the premise underlying this reviewer’s query because it has been known for over 45 years that 90% of intravenous dose of Fosfomycin is excreted in urine by 6h as opposed to only 20% of the oral Fosfomycin dose by 6h time point [Shimizu, K. Fosfomycin: Absorption and excretion. Chemotherapy 1977, 23 Suppl 1, 153-158;Cadorniga, R.; Diaz Fierros, M.; Olay, T. Pharmacokinetic study of fosfomycin and its bioavailability. Chemotherapy 1977, 23 Suppl 1, 159-174]. Since the delay in urinary MIC of oral Fosfomycin is empirically proven to be longer than the delay in urinary MIC of intravenous Fosfomycin, we were wondering about the scientific reasoning underpinning the reviewer’s query. In fact, the reviewer query is as baseless as the speculation that the delay in urinary MIC of Ertapenem (parenteral antibiotic) [Cunha et al 2016. Predictors of ertapenem therapeutic efficacy in the treatment of urinary tract infections (UTIs) in hospitalized adults: the importance of renal insufficiency and urinary pH. Eur J Clin Microbiol Infect Dis, 35, 673-679] among patients with renal insufficiency (creatinine clearance <50 mL/min) cannot be the basis for inferring the delay in urinary MIC of oral antibiotics in the threats section of SWOT.

Since only the absorbed drug can be excreted, bioavailability and the renal function status are two determinants in the renal excretion of Fosfomycin (oral and parenteral antibiotic) or Ertapenem (parenteral antibiotic). While bioavailability of intravenous Fosfomycin and Ertapenem is 100%, the bioavailability of oral Fosfomycin is 37%. Clearly, a lower and variable absorption from gastrointestinal tract is a factor in delaying the urinary MIC of oral Fosfomycin more than the intravenous Fosfomycin [Shimizu, K. Fosfomycin: Absorption and excretion. Chemotherapy 1977, 23 Suppl 1, 153-158;Cadorniga, R.; Diaz Fierros, M.; Olay, T. Pharmacokinetic study of fosfomycin and its bioavailability. Chemotherapy 1977, 23 Suppl 1, 159-174]. The significance of bioavailability in clinical outcomes was also underlined by a recent study by Mponponsuo et al 2023 from Canada. Study found that highly bioavailable antimicrobials at hospital discharge performed better in clinical outcomes than antimicrobials with lower bioavailability [ Mponponsuo et al Highly versus less bioavailable oral antibiotics in the treatment of gram-negative bloodstream infections: a propensity-matched cohort analysis. Clin Microbiol Infect 2023, 29, 490-497]. We have now cited this study in the review to underscore the significance of bioavailability and for adding pharmacokinetics to the continuing medical education of antibiotic prescribers, so they are not confused by the pharmacokinetic terms.

While the delay in urinary MIC will be higher in patients with renal insufficiency, the delay in urinary MIC for oral antibiotics is compounded by the delay in oral absorption. Therefore, we would like to reiterate that our review is focused on oral antimicrobials and the sources for the delay in urinary MIC stemming from the variability in kinetics of absorption, and excretion and the individual variability in bladder capacity.

Line 60 it’s good that you included gepotidacin as a novel oral antibiotic for UTI, but nowhere does these clinical trials discuss delay in urinary MIC after administration, where are you inferring this information?

Response: We thank the reviewer for appreciating the inclusion of gepotidacin. While the clinical trials are focused on the clinical efficacy of antimicrobial, we encourage the reviewer to read the reference 26 on the pharmacokinetic study on gepotidacin by Barth et al 2023 in healthy volunteers to understand the inferred delay in urinary MIC of human subjects with varying degrees of renal impairment and oral bioavailability. Unlike plasma MIC, the timepoints for urinary MIC are plotted as midpoints of urinary collection periods which itself are variable as illustrated by mirage graph of Fig. 3. As indicated in the detailed response to previous comment, the sources for the delay in urinary MIC stem from the variability in kinetics of absorption, and excretion and the individual variability in bladder capacity. The posited delay in urinary MIC is difficult to quantify due to challenges in sampling period and inter-individual variability in kidney function as evident from study by Edwina et al 2023 and Barth et al 2023. The challenges are visually displayed by the ‘window of opportunity” timeframe in Fig.3 of the root cause analysis,

Line 61 what is an asymmetric advantage?

Response: We were following the journal advice on expanding the reach of our review to a broad audience. Towards that we used the self-explanatory phrase “asymmetric advantage” at two places in the review to explain the advantage gained by uropathogens through AMR in their combat with antimicrobials and then to explain the advantage of safe dose escalation via intravesical administration to kill uropathogen, which is not possible with oral route. While advantage is synonymous with strength in SWOT, asymmetry refers to the unmatched strength of an opponent in a competition. AMR certainly confers an asymmetric advantage to uropathogens.  We may have overestimated the self-explanatory potential of the phrase “asymmetric advantage” but asymmetry of information about treatment between patients and clinicians is well known within health care community.

Antimicrobial resistance

Line 77-79 this reference does not match the sentence, what is a clinical governance?

Response: We beg to disagree with the reviewer remark, and we can only hope that reviewer is not judging the article (book) by its title or its abstract (cover). In the revised manuscript, we have added a recent Lancet reference emphasizing the importance of governance frameworks for tackling AMR. It is a widely accepted “clinical governance oversight of prescriber practices” is a feature of antimicrobial stewardship (AMS) in which antibiotic prescription behavior of clinicians is audited by public health professionals.

Line 86 Postulate is used incorrectly here.

Response: We thank the reviewer for noticing the erroneous usage, we have now corrected it to inference.

Line 90-92 needs a reference.

Response: We thank the reviewer for the suggestion, and we have now added reference.

Line 96-100 sentence should be reworded, it’s confusing to include both US and UK data in one sentence.

Response: We thank the reviewer for the suggestion and now US and UK data is now discussed separately.

Line 109-115 again this key study data is not correctly represented and this is another example of error in data evaluation.

 Response: We have noted the reviewer suggestion and to avoid any further confusion, the summary results on FQ-resistance prevalence are stated as reported by the authors in the cited study.

The urinary bladder

3.1 neutrophils – while this section is interesting, we failed to see the relevance of this section to the rest of the paper. Would just keep the first paragraph of this big section.

Response: We would like to keep this section as it is relevant for discussing the rationale for not giving antibiotics for asymptomatic bacteriuria (ASB) but exploring non-antibiotic approaches in the discussion section of manuscript.

Line 170-173 this statement doesn’t make sense, decline in chemokines after treating cystitis with oral antibiotics should indicate good resolution of infection, but are you saying this is a negative thing because of the lack of neutrophils?

 Response: We have now revised the sentence to improve clarity. Yes, reviewer clearly noted that the decline in urinary chemokines indicates good resolution of infection.

Cystitis Etiology and Treatment

Line 183-186 why discuss urothelial carcinoma? We see no relevance to the article.

Response: We beg to disagree with the reviewer’s statement as our intent here is to draw the attention of readers to the inverse relationship between the prevalence of cystitis and urothelial carcinoma in men and women. The gender-based differences in prevalence are relevant in understanding the pathogenesis of two disease and whether women accrue any anticancer benefit from being predisposed to cystitis or more frequent shedding of urothelium in delaying or retarding carcinogenesis.  In that context, we cited a recent Danish study suggesting the link between UTI and urothelial carcinoma, which deserves scrutiny of the scientific community (Pottegård et al 2020. Urinary tract infections and risk of squamous cell carcinoma bladder cancer: A Danish nationwide case-control study. Int J Cancer;146(7):1930-1936.)

Line 193-194 confusing.

Response: We thank the reviewer for the opportunity to clarify and we have now revised the sentence to avoid confusion.

Line 197 no reference.

Response: We have now added two new references to support the statement in revised manuscript.

Line 199-201 this needs a reference, imaging is usually not needed when diagnosing complicated vs uncomplicated UTI.

Response: We thank the reviewer for bringing this up and we have chosen to delete that statement in revised manuscript as the review is not focused on imaging.

Line 201-204 this sentence is confusing, why is the need for translational research important here? Are you stating that since mouse models cannot accurately predict human pathophysiology, we need more studies in pig bladder? But why is translational research important in the first place? There are many in vivo human studies of oral antibiotics and UTI.

Response: We thank the reviewer for raising this question on the need of translation research in cystitis treatment, which is certainly not the main topic of this review. We have chosen to remove the paragraph on translation research from the revised manuscript.

Line 224-227 can’t this inter-individual variability be seen in other routes of administration?

Response: We thank the reviewer for the comment, which we have already addressed in responses to earlier comments related to the inter-individual variability in pharmacokinetics of antimicrobials stemming from the unique attributes of oral route. Briefly stated, the prophylaxis and treatment of cystitis is limited to just three routes:  oral, intravenous and intravesical. Of these three routes, only oral route is uniquely characterized by inter-individual variability in absorption because the absorption is 100% by intravenous route [Shimizu, K. Fosfomycin: Absorption and excretion. Chemotherapy 1977, 23 Suppl 1, 153-158;Cadorniga, R.; Diaz Fierros, M.; Olay, T. Pharmacokinetic study of fosfomycin and its bioavailability. Chemotherapy 1977, 23 Suppl 1, 159-174] whereas absorption via intravesical treatment is redundant for killing uropathogens [Stalenhoef et al 2019. Intravesical Gentamicin Treatment for Recurrent Urinary Tract Infections Caused by Multidrug Resistant Bacteria. J Urol, 201, 549-555].

Line 257 this reference is for Ertapenem, which is a parenteral antibiotic, the authors cannot infer threats to oral antibiotic from this reference.

Response: We have partly addressed this scientifically unsound speculation of reviewer in response to initial comment on oral Fosfomycin as renal excretion of antimicrobials occurs irrespective of the route of administration whether enteral or parenteral. The sources of variability in renal excretion of antimicrobials do not vary with the route of administration whether enteral or parenteral, only the magnitude of variability increase with the oral administration of [Shimizu, K. Fosfomycin: Absorption and excretion. Chemotherapy 1977, 23 Suppl 1, 153-158;Cadorniga, R.; Diaz Fierros, M.; Olay, T. Pharmacokinetic study of fosfomycin and its bioavailability. Chemotherapy 1977, 23 Suppl 1, 159-174].

Given that both oral and parenteral antibiotics intended for cystitis are excreted by kidney into bladder, factors that compromise kidney function will also impact renal excretion of oral as well as parenteral antibiotics. Therefore, it is correct to infer threats to oral antibiotic from the variability in renal excretion of Ertapenem as also proven by the cited study on oral and intravenous administration of Fosfomycin, studied by Edwina et al 2023 and Shimizu 1977. We have now added multiple references on empirical evidence that contradict the scientifically baseless speculation of reviewer.

We are not aware of any scientific reasoning or empirical clinical evidence that forbids drawing the inference about the threats to oral antibiotics drawn from the renal excretion of parenteral antibiotics. Moreover, if one views this comment of the reviewer together with subsequent comments of the reviewer, authors can only reach the conclusion that the reviewer is prone to making unfounded assumptions and speculations or “made up” statement.

To understand the differences in oral and intravenous administration of the same antibiotic, Fosfomycin, we encourage the reviewer to read the cited study of Edwina et al 2023, especially figure 6-8 on plasma and urine levels of Fosfomycin after oral as well as intermittent and continuous intravenous infusion (parenteral) in healthy volunteers. Edwina et al 2023 studied the variability in kinetics of urinary MIC stemming from the variability in glomerular filtration rate (GFR). While the study by Edwina et al 2023 did not include human subjects with varying degrees of renal impairment, the product monograph of Fosfomycin states that renal impairment significantly decreases the percentage of Fosfomycin recovered in urine to 11% and a decline in creatinine clearance from 54.2 mL/min to 7.3 mL/min raises the plasma half -life of Fosfomycin from 11 hours to 50 hours.

Line 259 define infection ascent.

Response: We thank the opportunity to clarify, and we have now replaced infection ascent it with simpler terms of cystitis progressing to pyelonephritis and urosepsis, that is infection ascending up from lower urinary tract to upper urinary tract.

Line 261 overuse of antibiotics in general? Or just oral antibiotics. This reference refers to antibiotics in general.

Response: We thank the reviewer for insightful comment, and we have now replaced the references for oral antibiotics.

Line 265 weakness – this sentence is made up, and the references listed here do not support this statement.

Response: Authors can only hope that the reviewer studied the three references before making the stated erroneous judgment. For the benefit of the reviewer and the editorial board, we would like to categorically state that the statement on the weaknesses of oral antimicrobials in our manuscript is predicated on inter-individual variability in urine levels of antimicrobials impacting the efficacy in cystitis as reported by Sharma et al 2023 on oral nitrofurantoin and Zhanel et al 2020 on oral and intravenous Omadacycline and the review authored by Von Vietinghoff et al 2023 on physiochemical composition of urinary compartments.  

We briefly state the summary of three studies here that model simulations of Sharma et al 2023 on oral nitrofurantoin (NFT) “predicted that the liver of individuals with a moderate-to-severe glomerular filtration rate (GFR) is exposed to two-to-three-fold higher concentrations of NFT than individuals with a normal GFR, which coincided with a substantial reduction in the NFT urinary concentration” , which implies  lower efficacy (weakness) in cystitis. Zhanel et al 2020 study reported delayed oral absorption of omadacycline relative to its intravenous infusion causing the variability in terminal elimination half-life of 13.5-17.1 h and variability in total clearance of 8.8-10.6 L/h in healthy volunteers with the predicted variability even higher in individuals with cystitis. Von Vietinghoff et al 2023 discussed the emergence of AMR with OAT in light of the variability in urine flow and urine composition.

It is unclear to us how the reviewer concluded that the aforementioned studies do not support the sources of inter-individual variability in urine levels of antimicrobials ascribed as weakness in our thesis of SWOT. We concede that the three studies do not use the term ‘weakness” as a key word because we are the only one who are inferring inter-individual variability in plasma and urine levels of OAT as weakness for the purpose of SWOT analysis. To bolster our argument, we have now added more clinical references, which can only be understood by someone who does not vehemently oppose any discussion on the pharmacokinetics of antimicrobials.

Line 268 opportunities - we do have predictable oral bioavailability for most oral antibiotics, where is the reference to this unpredictability? The reference listed here does not discuss this unpredictability. This statement is mostly made-up as well, are you saying oral antibiotics with 100% bioavailability will not encounter AMR?

Response: Contrary to reviewer’s assertion, we do not have predictable oral bioavailability of antimicrobials in bed-ridden cystitis patients [Forsberg et al 2023,Bioavailability of Orally Administered Drugs in Critically Ill Patients. J Pharm Pract, 36, 967-979]. Moreover, oral bioavailability information included in product inserts of drugs is generally determined on healthy volunteers and not usually on cystitis patients, who may or may not be consuming food. Furthermore, oral bioavailability differs with race, food intake, inter-individual variability in the gastric emptying time, posture (supine or ambulatory) and general health status of the patient. There are plenty of reports on unpredictability of oral bioavailability of OAT and we have now added additional references on unpredictability [Stocco et al 2020. Pharmacogenomics of Antibiotics. Int J Mol Sci, 21;  Tolentino-Hernández  et al 2020. Oral Ciprofloxacin Pharmacokinetics in Healthy Mexican Volunteers and Other Populations: Is There Interethnic Variability? Arch Med Res.51(3):268-277; Zhanel et al 2022 , Sulopenem: An Intravenous and Oral Penem for the Treatment of Urinary Tract Infections Due to Multidrug-Resistant Bacteria. Drugs.82(5):533-557]

Once again, reviewer’s baseless assertion of “made up” statement is challenged by multiple references on oral bioavailability of OAT which is unlikely to approach the 100% threshold of intravenous bioavailability. Even if the oral bioavailability was predictable, it can still incur delay in urinary MIC and leave a window of opportunity for  AMR to emerge as illustrated in Fig. 3. A recent study by Mponponsuo et al 2023 from Canada compared the clinical outcomes with less and more bioavailable antimicrobials after hospital discharge and highly bioavailable performed better than less bioavailable ones.  

Line 277-288 this section is confusing to read, and the authors should take a closer look at reference 20 to summarize their PK/PD findings for nitrofurantoin, and why it’s on indicated for uncomplicated cystitis and not systemic infections. Reference 101 talks about lidocaine, which is not relevant to this section. Reference 102 is a murine study on pregnant mice, since you already discussed previously that murine models are not enough to predict human data, why include it here?

Response: After answering several scientifically flawed assertions of the reviewer, we can’t help but notice that the reviewer labels any manuscript text that is dense in pharmacokinetics as “confusing to read”. To rephrase Einstein’s quote, if pharmacokinetic ignorance is part of the problem of AMR then we cannot design solutions of AMR by ignoring pharmacokinetics or by labelling it “confusing to read”.

We wrote the section in 277-288 to highlight the clinical significance of the less understood, body weight dependent pharmacokinetic term, “volume of distribution” which is important to understand the differences in pharmacokinetics of antibiotics administered for treating cystitis in pregnant and non-pregnant women and infections in burn victims.

We have now rewritten that section after performing an in-depth study of reference 20 and using the published tables to plot the graph in Fig.2B. We also used PK/PD of nitrofurantoin to explain why just 20-25% absorbed dose of nitrofurantoin excreted in urine is enough to treat uncomplicated cystitis, whereas remainder of 75-80% of absorbed dose of nitrofurantoin NOT excreted in urine fails to generate high enough plasma levels to counter systemic infections. In that context, the pregnant mice study is discussed to illustrate the effect of pregnancy mediated rise in volume of distribution affecting the pharmacokinetics of nitrofurantoin. The volume of distribution is also relevant in understanding the differences in clearance of antimicrobials in burn victims.

We understand that familiarity with standard pharmacology terms can be variable in the clinical community. However, understanding the concept of volume of distribution is critical for understanding the dilution and rapid clearance of 75-80% of absorbed dose of nitrofurantoin. We refer the reader and reviewer to reference 101 for further information on large volume of distribution causing the confusion in lidocaine pharmacokinetics.

Line 293 “errors in treatment of cystitis” not sure what this means. Are you saying we are not effectively treating cystitis? Where is the reference for this?

Response: We have now revised the statement to improve the clarity and added references on treatment failures with oral antimicrobial treatment. Yes, there is plenty of literature supporting the proposition that cystitis is not being treated effectively. Our review was triggered by OAT of cystitis being part of the problem in AMR emergence. In that context, the review discusses published papers on antibiotic and non-antibiotic approaches as well as treating the pain symptoms of cystitis with NSAIDs and let neutrophils tackle the uropathogens to provide symptom relief without driving the emergence of AMR.

Line 205 “toxicity and lower therapeutic index” what does this mean?

Response: As stated in response to previous comments, “volume of distribution”, toxicity and therapeutic index are standard pharmacology terms. A drug with lower or narrow therapeutic index is characterized by a narrow margin between toxic dose and effective dose, which prevents safe dose escalation for eradicating uropathogens. While narrow therapeutic index of gentamicin prevents its dose escalation by intravenous route (100% bioavailability) for killing uropathogens in cystitis, lower bioavailability of intravesical route affords dose escalation of gentamicin for safe killing of gentamicin.

Line 304 this statement is disingenuous and misleading since the reference only demonstrated that Fosfomycin and polymyxin needs this much higher of a concentration.

Response: We again disagree with the reviewer’s remark and perhaps reviewer missed the lower range of the interval [4-1000fold] that indicates the vast range of antimicrobials discussed in the said reference. Our review clearly states the range of 4-1000fold to cover the mutant prevention concentration of beta lactams ≥ 4 of MIC; aminoglycoside ≥ 20 of MIC; fluoroquinolones ≥ 35 of MIC; tetracyclines, ≥ 50 of MIC; polymyxin B, ≥ 808 of MIC; and fosfomycin, ≥ 3136 of MIC.

Line 307 “military doctrine”?

Response: We have now revised the statement by deleting the superfluous phrase.

Figure 3 – what is the unit for Time here? What do you mean “mirage” and “unknowledge and indeterminable” real world data points? Is this graph accurate at all?

Response: We thank the reviewer for giving us the opportunity to clarify. This is a graph for illustrative purposes only, displaying the non-zero interval in hours for the initial below MIC exposure in urine of uropathogens that gives rise to progenies with progressively higher tolerance in successive generations to higher concentrations of antimicrobials arriving in urine. The exact duration of non-zero interval displayed in Fig. 3 is determined by a multitude of factors including variable oral bioavailability, renal function status, urine flow rate and pre-existing urine volume before the start of drug excretion. The window of opportunity displayed in the graph is consistent with the recovery of only 20% of the oral dose of Fosfomycin in urine by 6h after oral administration as opposed to 90% recovery of intravenously injected dose of Fosfomycin by 6h. Since the exact duration of non-zero interval displayed in Fig. 3 is a function of the indeterminable kinetics of the orally absorbed antimicrobial’s arrival in stored urine, the non-zero interval also merits the moniker, “mirage”.

If the reviewer is still not convinced with our reasoning for displaying the graph, we are open to deleting Fig.3, but we believe it serves as useful prop to explain the root cause analysis and asymmetric advantage of intravesical administration of antimicrobials to counter AMR.

Line 330 “window of opportunity” do you have reference to back up this claim of development of resistance after first dose? Reference 107-108 discuss resistance but not in this context.

Response: We thank the reviewer for giving us the opportunity to clarify. Basic research by Wright et al and Gonze et al (reference 5-7 of the review) have shown that dynamic bacterial communities are extremely sensitive to the initial conditions in AMR development. Therefore, the non-zero interval of below MIC exposure after oral dosing creates those initial conditions in bladder that are favorable for uropathogens to mutate and gain AMR. The time frame needed for uropathogens to mutate in vitro is comparable to the time frame that uropathogen get for below MIC exposure in urine after oral dosing of OAT. According, there is scientific basis for connecting the dots between basic and clinical research in illustrative plot of Fig.3 to make a plausible hypothesis for a delay in urinary MIC triggering the emergence of AMR in patients predisposed to recurrent infections.

Line 335-336, the reference listed here does not match the statement.

Response: Once again, we disagree with the speculative remark of the reviewer. We would like to clarify that the review is not simply a compilation of statements made in the title of the research article but a comprehensive analysis of facts in the referenced studies. The statement in question pertains to the relative cardiac output percentage received by kidney and bladder wall, respectively which can be empirically determined from the percent dose of injected radiolabeled probe or paramagnetic dye measured in bladder wall or kidney after injection.

Line 363 why talk about dog models? Did the authors focus on pig bladder models earlier? This data is also contradictory to the core hypothesis of this paper, since in some of the dogs they found faster Tmax in urine than serum concentrations.

 Response: We highly appreciate the insightful comment of the reviewer and have now chosen to delete the dog study from the review. As reviewer may appreciate that it can be difficult to get urine samples “on demand” from dogs and that sampling flaw may have generated the distorted inference of urinary Tmax  occurring an hour earlier than plasma Tmax.

Discussion

Line 382-389 not sure why this paragraph is needed here.

Response: We thank the reviewer for the opportunity to clarify because the paragraph is needed as a prologue to the discussion.

Line 391 “tortuous”?

Response: We have deleted the superfluous word

Line 393-395 which experts? This reference is from 1979 and maybe not as relevant to modern practice. Can you find a more recent reference to this expert opinion on healthy voiding?

Response: We have added two new references to support the statement.

Line 397 the authors have not defined ASB yet, and asymptomatic bacteriuria is an important stewardship topic, if the authors want to focus on this instead, would recommend reading guidelines and discuss why it is not recommended to treat ASB.

Response: We thank the reviewer for pointing that out. ASB is now clearly defined in discussion and in neutrophils section. Yes, ASB is an important AMS topic and we have clarified that in the discussion and ASB is discussed in relation to non-antibiotic approaches for prophylaxis of cystitis.

Line 399-401 would reword this sentence, again data from other countries should not be used to infer US data.

Response: We thank the reviewer for the comment and the statement is now revised.

Line 403-405 this was basically the same sentence from line 96-100.

Response: We have now deleted the sentence from discussion and leaving it in introduction.

Line 421-469 this was the same Root Cause analysis from before, but just moved to a different section. This section also remained mostly unchanged. This also didn’t fully address our initial edits. Again, are you saying intravesical antibiotics are better and should be given to all patients with UTIs, instead of oral antibiotics?

Response: We thank the reviewer for the opportunity to clarify. We have made every attempt to address the reviewer comments to every version and text has been moved around for coherent flow after extensive changes prompted by numerous comments of the reviewers. With regard to the initial edits of reviewer, we think we have clearly addressed them because in response to our original manuscript, this reviewer made the unfounded speculation, “that intravesical antibiotics are outdated” despite numerous clinical studies in the recent past. That irrational speculation has now morphed into an irrational rhetoric on whether intravesical are better than oral antibiotics.

While intravesical antibiotics are not better than oral antibiotics for all outpatients, but in the era of personalized medicine, where treatment for UTI is highly individualized, intravesical antibiotics may be better than oral antibiotics for hospitalized patients, patients with renal insufficiency, immunocompromised and neurologic patients. For such individuals, intravesical antibiotics may comply with the principle of 5Ds to counter the emergence of AMR as noted by multiple studies. Moreover, intravesical route offers an asymmetric advantage of safe dose escalation to counter uropathogens endowed with AMR.  Likewise, oral antibiotics are not advisable for asymptomatic bacteriuria. However, even if every physician makes guidelines concordant OAT prescription and follows best AMS practices for cystitis, there may be physiological basis for AMR persistence and recurrence of infection in some individuals as illustrated by Fig.3.

 Line 479-482 most guidelines already identify cystitis as its own infection and should be treated differently from other complicated UTIs.

Response: We are glad that finally, we are on the same side of argument.

Line 508 where is the actual reference for this? Dequalinium chloride is not routinely used and is mostly indicated for throat infections, how is this relevant to UTI?

Response: We would like to clarify that this section is forward looking and discusses the potential approaches to slow the pace of AMR development. Since Escherichia coli is the most common uropathogen and fluoroquinolones are used against uropathogen, we foresee a day when Dequalinium chloride is also administered for cystitis.

Line 515-518 not sure what you are trying to say here, guidelines are constantly changing based on new clinical data and new antibiotics.

Response: We intended to say here is that safe antimicrobial drugs cannot be brought to the market at the same pace as the pace of AMR development in uropathogen. Therefore, there is need to appreciate the factors responsible for AMR development with OAT and appreciate the role of pharmacokinetics in clinical outcomes.